# vCache: Verified Semantic Prompt Caching

**Luis Gaspar Schroeder**[1,2]    **Aditya Desai**[1]    **Alejandro Cuadron**[1,3]    **Kyle Chu**[1]
**Shu Liu**[1]    **Mark Zhao**[4]    **Stephan Krusche**[2]    **Alfons Kemper**[2]    **Matei Zaharia**[1]
**Joseph E. Gonzalez**[1]

[1]University of California, Berkeley    [2]Technical University of Munich    [3]ETH Zurich
[4]Stanford University

## Abstract

Semantic caches return cached responses for semantically similar prompts to reduce LLM inference latency and cost. They embed cached prompts and store them alongside their response in a vector database. Embedding similarity metrics assign a numerical score to quantify the similarity between a request and its nearest neighbor prompt from the cache. Existing systems use the same static similarity threshold across all requests to determine whether two prompts can share similar responses. However, we observe that static thresholds do not give formal correctness guarantees, result in unexpected error rates, and lead to suboptimal cache hit rates. This paper proposes vCache, the first verified semantic cache with user-defined error rate guarantees for predictable performance. It employs an online learning algorithm to estimate an optimal threshold for each cached prompt, enabling reliable cache responses without additional training. Our experiments show that vCache consistently meets the specified error bounds while outperforming state-of-the-art static-threshold and fine-tuned embedding baselines with up to $12.5\times$ higher cache hit and $26\times$ lower error rates. We release the vCache implementation and four benchmarks to support future research.

## 1    Introduction

Large language models (LLMs) power applications ranging from conversational assistants to search engines and code generation, but their widespread use is limited by the high computational cost and inference latency (Zhao et al., 2023; Xiong et al., 2024; Achiam et al., 2023). Each new prompt requires multiple expensive forward passes through the model, which makes deployments costly and slow (Kwon et al., 2023). Prompt caching offers a natural way to mitigate this issue: if a prompt has already been answered, the system can return the cached response instead of performing another inference. Traditional exact string-match caching reduces cost by returning responses for repeated prompts, but it fails whenever the same intent is expressed in different words (Zhu et al., 2024). For example, a cache that already answered "Which city is Canada's capital?" should also return the same response when later asked "What is the capital of Canada?". Semantic caching addresses this limitation by retrieving responses for prompts that are semantically similar, even if their lexical form differs, and reduces inference latency by up to 100× (Bang, 2023). Semantic caches are effective in single-turn interactions with short to medium context, such as web search queries or classification tasks, where requests reappear in paraphrased forms but map to the same underlying response (Liu et al., 2024b; Wang et al., 2024). In this paper, we study the reliability of semantic caches in returning correct responses for semantically similar prompts.

Semantic caches operate as follows. The cache embeds every prompt request $x$ into a vector $\mathcal{E}(x) \in \mathbb{R}^d$ and retrieves the semantically most similar prompt $nn(x)$, alongside its response $r(nn(x))$, from a vector database (Pan et al., 2024). The cache measures similarity (e.g., cosine similarity) between two embeddings using $s(x) = sim\big(\mathcal{E}(x), \mathcal{E}(y)\big) \in [0, 1]$. If no sufficiently similar prompt is found, an LLM generates a response and adds the embedded prompt along with the response to the vector database in the cache.

To determine whether a new prompt is sufficiently close to an existing prompt in the cache, state-of-the-art semantic caches rely on a user-selected threshold $t \in [0, 1]$ (Bang, 2023; Li et al., 2024;

Dasgupta et al., 2025; Razi et al., 2024; Sudarsan & MasayaNishimaki, 2024). If $s(x) \geq t$, the system performs exploitation (cache hit) by returning the cached response $r(\text{nn}(x))$. Otherwise, it performs exploration (cache miss) by querying the model for a new response $r(x)$. The cache adds $\mathcal{E}(x)$ to the vector database, stores $r(x)$ in its metadata, and returns $r(x)$.

However, selecting an appropriate threshold $t$ is nontrivial. If the threshold $t$ is set too low, the system may treat unrelated prompts as similar, resulting in cache hits where the retrieved response $r(\text{nn}(x))$ differs from the correct output $r(x)$. These false positives reduce response quality and compromise cache reliability. If $t$ is too high, the system may forgo correct cache hit opportunities and invoke the model unnecessarily (Rekabsaz et al., 2017).

Existing systems use the same static similarity threshold across all requests. Users either use a predefined threshold (e.g., 0.8) or determine one by testing multiple values upfront (Dasgupta et al., 2025; Li et al., 2024; Dan Lepow, 2025; Razi et al., 2024; lit, 2025; Bang, 2023). This approach assumes that similarity correlates uniformly with correctness across all prompts and their embeddings. However, two prompts may be close in embedding space yet require different responses. Figure 3 illustrates that correct and incorrect cache hits have highly overlapping similarity distributions, suggesting that fixed thresholds are either unreliable or must be set extremely high to avoid errors, making them suboptimal. Another significant limitation of existing semantic caches is the lack of error-rate guarantees. While the latency benefits of caching are appealing, the risk of returning incorrect responses can outweigh those advantages. For widespread adoption, semantic caches must adhere to user-defined error rate tolerances.

We propose vCache, the first verified semantic cache with theoretical correctness guarantees. vCache learns a separate threshold (Figure 1) for each embedding in the cache, capturing the threshold variability observed in Figure 3. It requires no upfront training, is agnostic to the underlying embedding model, and dynamically adapts its thresholds to the data distribution it encounters. As a consequence, vCache is robust to out-of-distribution inputs. To our knowledge, no prior work in semantic caching 1) learns thresholds in an online manner and 2) guarantees their correctness.

We adopt a probabilistic framework to bound the error rate conditioned on a learned per-embedding threshold. When deploying vCache, the user specifies a maximum error rate bound $\delta$, and the system maximizes the cache hit rate subject to this correctness constraint (Figure 2). Let $\mathbf{vCache}(x)$ denote the response returned by vCache. Let $\tau$ denote the exploration probability—a value monotonically decreasing in the likelihood of being correct—and calibrated such that the overall error rate remains below the user-specified bound (Section 4). The decision rule modeling the probability of being correct for whether to exploit the cached response or explore an LLM inference is given by:

$$\mathbf{vCache(x)} = \begin{cases} r(\text{nn}(x)) & \text{Uniform}(0,1) > \tau, \\ \text{LLM}(x) & \text{otherwise.} \end{cases} \tag{1}$$

An illustrative overview of the vCache workflow and system architecture is provided in Appendix B.

We evaluate the effectiveness of vCache in terms of correctness guarantees and overall performance. To assess generalizability, we compare vCache across three embedding models, two LLMs, and five datasets. We find that vCache consistently meets the error-rate bounds and outperforms static threshold baselines, even when using fine-tuned embeddings. Specifically, it achieves up to **12.5×** **higher cache hit rates** and **reduces error rates by up to 26×**. Our main contributions are as follows:

1. We propose **vCache**, the first predictable semantic cache that enforces a **user-defined correctness guarantee** by bounding the error rate.
2. We introduce an **online threshold learning algorithm** that requires no prior supervised training, adapts to the observed data distribution, and is agnostic to the choice of embedding model.
3. We demonstrate that **embedding-specific, dynamic thresholds** improve decision quality. By learning a separate threshold per cached embedding, vCache achieves equal or better performance compared to both static thresholding and fine-tuned embeddings.
4. We **introduce four benchmarks**, derived from five real-world datasets, that capture common semantic cache use cases: classification tasks, search queries, and open-ended prompt distributions.
5. We **release the vCache implementation**[1] and **four benchmarks**[2] to support future research.

---

[1] https://github.com/vcache-project/vCache
[2] https://huggingface.co/vCache

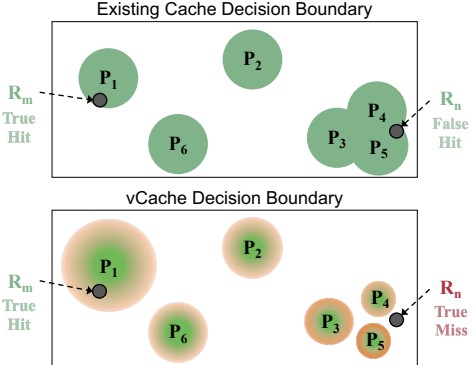

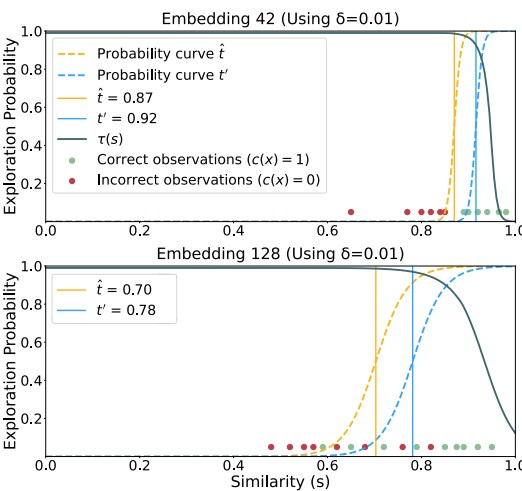

Figure 1: The static threshold in existing semantic caches enforces naive decision boundaries, resulting in either low cache hit or high error rates. vCache's embedding-specific and dynamic thresholds learn decision boundaries to guarantee a user-defined maximum error rate. Gradient shading reflects decreasing confidence in correctness as similarity to the cached embedding decreases.

Figure 2: Exploration probability for $emb_{42}$ and $emb_{128}$. Top: Observations are perfectly separable. Bottom: Observations are overlapping. vCache selects the optimal $\hat{t}$ and adjusts the exploration probability based on the user-defined $\delta = 0.01$.

## 2 RELATED WORK

Existing semantic caches, such as GPTCache (Bang, 2023) and industry variants (Razi et al., 2024; Dan Lepow, 2025; lit, 2025; Dasgupta et al., 2025; Li et al., 2024), use a global similarity threshold to make cache hit decisions. This assumes a uniform correlation between similarity and correctness across all prompts and embeddings. However, as illustrated in Figure 3, similarity distributions vary widely, making fixed thresholds unreliable. Further details are provided in Appendix F.9.

**Semantic Cache Optimization.** The threshold dilemma illustrated in Figure 3 can be addressed via two approaches: optimizing the embedding space or learning more effective thresholding strategies.

*Embedding Fine-tuning:* Zhu et al. (2024) propose a distillation-based method that fine-tunes embeddings for semantic caching, improving alignment between semantically equivalent prompts and their responses. A related challenge arises in image retrieval, where systems must determine whether a nearest neighbor corresponds to the correct target class. Zhang et al. (2023a) address this by introducing the Threshold-Consistent Margin loss, which enforces tighter intra-class cohesion and clearer inter-class separation by selectively penalizing negative pairs. However, they require supervised training, are limited to open-source embedding models, and can fail to generalize to out-of-distribution data at inference time (Hajipour et al., 2024). vCache's online learning algorithm does not require training, is model-agnostic, and generalizes to out-of-distribution data (see Appendix H.1).

*Threshold Optimization:* Threshold optimization learns a decision boundary over existing embeddings without modifying the embedding model itself (Zhang et al., 2023b). To our knowledge, no prior work in semantic caching learns thresholds online at inference time. Yet, as shown in Figure 3, the optimal similarity threshold varies significantly across embeddings, motivating embedding-specific and online threshold estimation. Related ideas have been explored in incremental learning. For example, Rudd et al. (2017) propose the Extreme Value Machine (EVM), which models class boundaries using extreme value theory to support generalization to unseen categories. However, these methods do not guarantee user-defined error rates. In contrast, we introduce the first online algorithm that estimates per-embedding thresholds for semantic caches while satisfying a user-defined error bound.

**Semantic Cache Guarantees.** Even with high-quality embeddings and a presumably carefully tuned threshold, semantic caches remain inherently approximate. Unless a threshold of 1.0 is used (effectively restricting cache hits to exact prompt matches), there is always a risk of returning incorrect responses (Razi et al., 2024; Jim Allen Wallace, 2024). Despite this, existing semantic caching systems rely on fixed thresholds to decide whether to return a cached response (Dasgupta

et al., 2025; Li et al., 2024; Dan Lepow, 2025; Razi et al., 2024; lit, 2025; Bang, 2023). As a result, they offer no formal guarantees on accuracy or error rates, making it difficult to justify their reliability in production environments. To address this, we propose vCache, the first semantic caching system that combines competitive performance with a user-defined correctness guarantee.

## 3 OVERVIEW OF SEMANTIC CACHING

Let $\{x_1, x_2, \ldots, x_n\}$ be the set of all prompts inserted into the cache, in that order. Note that this set excludes prompts for which a cache hit was found and served. Let $\mathcal{D}$ denote all the data stored in the cache. For each prompt $x$ inserted into the cache, we store its vector embedding $\mathcal{E}(x) \in \mathbb{R}^d$, the true response $r(x) = \text{LLM}(x)$ produced by the LLM, and optional additional metadata $\mathcal{O}(x)$. Given $\mathcal{E}(x)$, the cache retrieves the most similar prompt from the vector database with an approximate nearest neighbor search (Arya et al., 1998), where

$$\text{nn}(x) = \arg\max_{y \in C} sim\big(\mathcal{E}(x), \mathcal{E}(y)\big). \tag{2}$$

In vCache, for a prompt $x_i$, the metadata $\mathcal{O}(x_i)$ stores similarity and response match information for all future prompts $x_j$ (with $j > i$) such that $\text{nn}(x_j) = x_i$. The exact workings of vCache are presented in Algorithm 1. The sets $\mathcal{D}$ and $\mathcal{O}$ can be represented as follows:

$$\mathcal{D} = \Big\{ \big(\mathcal{E}(x_i), r(x_i), \mathcal{O}(x_i)\big) \Big\}_{i=0}^{n} \qquad \mathcal{O}(x_i) = \Big\{ \big(s(x_j), c(x_j)\big) \mid \text{nn}(x_j) = x_i \Big\}_{j=i+1}^{n} \tag{3}$$

where $s(x) \in [0, 1]$ is the similarity between $x$ and its nearest neighbor $\text{nn}(x)$ and $c(x)$ indicates if the cached response of $\text{nn}(x)$, $r(\text{nn}(x))$, matches the true response $r(x)$.

$$s(x) = sim(\mathcal{E}(x), \mathcal{E}(\text{nn}(x))) \qquad c(x) = \begin{cases} 1 & \text{if } r\big(nn(x)\big) = r(x), \\ 0 & \text{otherwise.} \end{cases} \tag{4}$$

**Algorithm.** Given a prompt, say $x$, we first compute the embedding $\mathcal{E}(x)$ and find its nearest neighbor $\text{nn}(x)$. The caching policy $\mathcal{P}$ then determines whether we should use the cached response for this prompt (exploit) or run the LLM inference (explore). In case we decide to exploit, $r(\text{nn}(x))$ is returned. Otherwise, we run $r(x) = \text{LLM}(x)$, compute $s(x)$ and $c(x)$, update the observations $\mathcal{O}(\text{nn}(x))$ and add $x$ to the database $\mathcal{D}$ using,

$$\mathcal{O}(\text{nn}(x)) = \mathcal{O}(\text{nn}(x)) \cup \{s(x), c(x)\}, \qquad \mathcal{D} = \mathcal{D} \cup \{(\mathcal{E}(x), r(x), \emptyset)\}. \tag{5}$$

The key challenge lies in designing a decision policy $\mathcal{P}(\ldots)$, as it directly impacts both the cache hit rate and the overall error rate.

**Policy of existing systems ($\mathcal{P}_{\mathbf{gptCache}}(\mathbf{s(x)})$).** In existing semantic caching systems, the decision function is implemented as a fixed threshold rule. Given a user-defined threshold $t$, the cache exploits if $s(x) \geq t$ by returning the cached response $r(\text{nn}(x))$. Otherwise, it explores by invoking the model for a response. As discussed in Section 2, this approach lacks formal guarantees and does not adapt to variation in similarity value distributions.

**Policy of vCache ($\mathcal{P}_{\mathbf{vCache}}(\mathbf{s(x)}, \mathcal{O}(\text{nn}(\mathbf{x})), \delta)$).** vCache replaces static thresholding with an embedding-specific decision function that respects a user-defined error bound $\delta$. If the function returns exploit, the cache is sufficiently confident and outputs the cached response $r(\text{nn}(x))$ (Algorithm 1, L5). Otherwise, it returns explore by inferring the LLM. Section 4 provides further details.

**Scope of definitions.** All quantities such as $nn(x)$, $s(x)$, and $c(x)$, are evaluated at a fixed (but arbitrary) point in time. Since the analysis is performed online, these definitions apply consistently across time steps. Policy discussion always refers to a specific embedding in the cache, with all parameters and estimates interpreted as conditional on it.

For ease of reference, a glossary of all symbols and functions is provided in Appendix A.

## 4 VCACHE

Given a user-defined maximum error rate $\delta$, vCache maximizes the cache hit rate while ensuring the probability of correctness remains above $1 - \delta$. Instead of relying on an unreliable static threshold or

**Algorithm 1** vCache Workflow

1: $e_x \leftarrow \mathcal{E}(x)$
2: $y \leftarrow \text{nn}(x)$
3: $s(x) \leftarrow sim(e_x, \mathcal{E}(y))$
4: **if** $\mathcal{P}(s, \mathcal{O}(y), \delta) = \text{exploit}$ **then**
5:    **return** $r(y)$
6: **else**
7:    $r(x) \leftarrow \text{LLM}(x)$
8:    $c(x) = \mathbf{1}(r(x) = r(y))$
9:    $\mathcal{O}(y) \leftarrow \mathcal{O}(y) \cup \{(s(x), c(x))\}$
10:   **if** $\neg c(x)$ **then**
11:      $\mathcal{D} \leftarrow \mathcal{D} \cup \{(\mathcal{E}(x), r(x), \emptyset)\}$
12:   **end if**
13:   **return** $r(x)$
14: **end if**

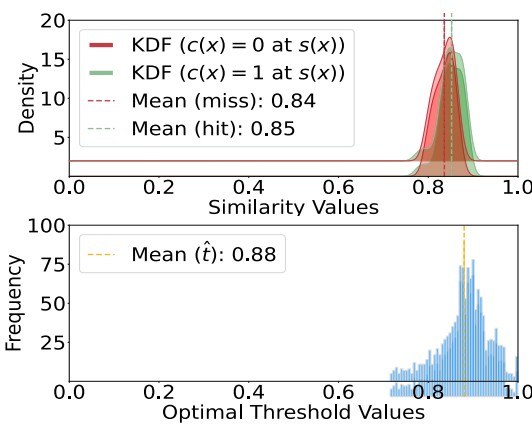

vCache workflow for deciding whether to exploit a cached response (cache hit) or explore an LLM inference (cache miss). The decision relies on the $\mathcal{P}_{\text{vCache}}$ policy (Section 4) and guarantees a user-defined error rate bound $\delta$.

Figure 3: Results from 45k samples in the Sem-CacheClassification benchmark. Motivates the need for dynamic, embedding-specific thresholds. Top: Similarity values of correct and incorrect explorations exhibit highly overlapping distributions. Bottom: Optimal per-embedding thresholds vary substantially, indicating that no single threshold can suffice across embeddings (see Appendix F.9).

fine-tuned embeddings, vCache models probability distributions to make cache hit or miss decisions. The distributions are specific to each embedding in the cache and model the probability of correct cache hits for a given similarity value. To remain dataset agnostic and avoid costly offline training, vCache estimates these distributions online by selectively generating labels for uncertain similarity values. Since generating a label requires an LLM inference, equivalent in cost to not using a cache, vCache minimizes such inferences. We refer to such inferences as explore and classify them as a cache miss. For a given prompt $x$, the cached response $r(\text{nn}(x))$ is considered uncertain when the observations $\mathcal{O}(nn(x))$ do not provide sufficient evidence to determine whether the cached response is correct ($c(x) = 1$). In contrast, if $\mathcal{O}(nn(x))$ provides sufficient evidence, vCache proceeds with exploit by returning the cached response $r(\text{nn}(x))$ without an LLM inference. The rest of this section formalizes these ideas and provides a detailed explanation of the vCache policy, $\mathcal{P}_{\text{vCache}}$.

### 4.1 USER GUARANTEE

One of the key features of vCache is that it takes a user-defined maximum error rate, $\delta$, and ensures that the overall performance of the cache remains within this error bound. Let $\mathbf{vCache}(x)$ denote the response returned by vCache, regardless of whether the decision was to explore or exploit. Then, an error rate guarantee of $\delta$ implies:

**Definition 4.1** (user-guarantee). *An error-rate guarantee of $\delta$ for vCache implies that the marginal probability of vCache returning the correct answer, given any arbitrary prompt $x$, is lower bounded by $(1 - \delta)$. In other words,*

$$\mathbf{Pr}(\mathbf{vCache}(x) = r(x)) \geq (1 - \delta). \tag{6}$$

To achieve the error guarantee, vCache probabilistically decides when to explore and when to exploit. Let $\mathbf{Pr}_{\text{explore}}(x|\mathcal{D})$ be the probability that, given a prompt $x$ and having accumulated data $\mathcal{D}$, vCache decides to explore. Then, we can decompose the probability that vCache is correct as,

$$\mathbf{Pr}(\mathbf{vCache}(x) = r(x)) = \mathbf{Pr}(\text{explore}|x, \mathcal{D}) + (1 - \mathbf{Pr}(\text{explore}|x, \mathcal{D}))\mathbf{Pr}(c(x) = 1|x, \mathcal{D}). \tag{7}$$

This expression reflects two disjoint events. First, vCache decides to explore with probability $\mathbf{Pr}_{\text{explore}}(x|\mathcal{D})$, and in this case, the output $\mathbf{vCache}(x)$ is same as $\text{LLM}(x)$ by design. In the second case, the vCache decides to exploit with probability $(1 - \mathbf{Pr}(\text{explore}|x, \mathcal{D}))$ and in this case, the probability of vCache being correct is represented as $\mathbf{Pr}(c(x) = 1|x, \mathcal{D})$ using notation from the

previous section. To ensure error guarantees are maintained, we should have,

$$\mathbf{Pr}(\text{explore}|x, \mathcal{D}) \geq \frac{(1-\delta) - \mathbf{Pr}(c(x) = 1|x, \mathcal{D})}{1 - \mathbf{Pr}(c(x) = 1|x, \mathcal{D})} = \tau_{\text{nn}(x)}(s(x)). \tag{8}$$

To meet the guarantees, vCache models the $\mathbf{Pr}(c(x) = 1|x, \mathcal{D})$. This inequality provides an actionable constraint: if the estimated probability of correctness from the cache is high, the system may exploit; if the estimate is low, the system must explore. As long as $\mathbf{Pr}(\text{explore}|x, D)$ is larger than the $\tau_{\text{nn}(x)}(s(x))$, the guarantees are achieved. Notation of $\tau$ is chosen to emphasize that it is a function over similarities and is specific to each embedding in the cache.

Since vCache can only estimate $\mathbf{Pr}(c(x) = 1 \mid x, \mathcal{D})$ based on a limited number of samples, it accounts for the uncertainty in the estimation by considering the confidence band of $\mathbf{Pr}(c(x) = 1 \mid x, \mathcal{D})$. The modeling details and the vCache policy are presented in the following subsection.

## 4.2 vCache Modeling

vCache imposes a sigmoid parametric model on the relationship between similarity and correctness. Specifically, for an arbitrary prompt $x$, the probability of correct cache hit is defined as

$$\mathbf{Pr}(c(x) = 1|x, \mathcal{D}) = \mathcal{L}(s(x), t, \gamma) = \frac{1}{1 + e^{-\gamma(s(x)-t)}}, \tag{9}$$

where $s(x) \in [0, 1]$ is the similarity of $x$ with its near neighbour. $t \in [0, 1]$ is an embedding-specific decision boundary parameter, and $\gamma > 0$ is a parameter controlling the steepness of the function. The sigmoid form is well-suited for this task: it induces a smooth and monotonic relationship between similarity and correctness probability and enables efficient maximum-likelihood estimation (MLE) of the threshold $t$ from labeled data (justification in Appendix E). By fitting a continuous likelihood function rather than enforcing a hard threshold, vCache generalizes better from limited observations.

The MLE estimates for the parameters, say $\hat{t}_{\text{nn}(x)}$ and $\hat{\gamma}_{\text{nn}(x)}$, using all the meta-data $\mathcal{O}_{\text{nn}(x)}$ solves the binary cross entropy loss,

$$\hat{t}_{\text{nn}(x)}, \hat{\gamma}_{\text{nn}(x)} = \arg\min_{t, \gamma} \mathbb{E}_{(s,c) \in \mathcal{O}_{nn(x)}} \left[ \Big(c \cdot \log(\mathcal{L}(s, t, \gamma))\Big) + \Big((1-c) \cdot \log(1 - \mathcal{L}(s, t, \gamma))\Big) \right] \tag{10}$$

Note that these parameters belong to a specific embedding in the cache (specifically $\text{nn}(x)$).

Since these estimates are based on a limited number of samples, estimating the true $\mathbf{Pr}(c(x)=1|x, \mathcal{D})$, and thus the correct $\tau_{\text{nn}(x)}(s(x))$ is not possible. To ensure we still achieve guarantees, vCache, instead computes a upper bound, say $\hat{\tau}$ for $\tau_{\text{nn}(x)}(s(x))$ using pessimistic values for $t, \gamma$ from the $(1 - \epsilon)$ confidence band for these points for some $\epsilon \in (0, 1)$. Let these estimates be $t'(\epsilon), \gamma'(\epsilon)$. We compute $\hat{\tau}$ using,

$$\hat{\tau} = \min_{\epsilon \in (0,1)} \frac{(1-\delta) - (1-\epsilon)\mathcal{L}(s(x), t'(\epsilon), \gamma'(\epsilon))}{1 - (1-\epsilon)\mathcal{L}(s(x), t'(\epsilon), \gamma'(\epsilon))} \geq \tau_{\text{nn}(x)}(s(x)). \tag{11}$$

The details of why $\hat{\tau} \geq \tau_{\text{nn}(x)}(s(x))$ and how to obtain confidence bands for $t$ and $\gamma$ are explained in Appendix C. Once we have $\hat{\tau}$, we have to ensure that the probability of exploration is above this value (see Eq 8 ). This is achieved by sampling a uniform random variable $u \sim \text{Uniform}(0, 1)$. If $u \leq \hat{\tau}$, the vCache explores, i.e., runs the LLM model to obtain the correct response. Otherwise, it exploits the cache by returning $r(\text{nn}(x))$. This randomized policy ensures that, in expectation, the system explores sufficiently often to meet the correctness guarantee while maximizing cache usage when reliability is high. The exact algorithm of how explore and exploit decisions are made is presented in Algorithm 2. Figure 2 illustrates the vCache modeling, where each subplot shows one cached embedding. Green and red points indicate correct and incorrect responses to observed similarities. The yellow dashed curve is the sigmoid model, with threshold $\hat{t}$ obtained by MLE. The blue dashed curve represents the sigmoid fit based on confidence bounds, with threshold $t'$ selected to meet the user-defined error bound $\delta$. The dark green curve $\tau(s)$ denotes the exploration probability, where for a given similarity $s$, vCache explores with probability $\tau(s)$ and exploits the cache otherwise.

## 4.3 vCache Algorithm

To summarize, the final vCache algorithm works as follows. First, for each incoming prompt $x$, it retrieves its nearest cached embedding $y = \text{nn}(x)$ and fits a logistic decision boundary $\hat{t}(y)$ using

---

**Algorithm 2** vCache Policy $\mathcal{P}_{vCache}(s(x), \mathcal{O}(\text{nn}(x)), \delta)$

1: **function** $\mathcal{P}_{vCache}(s(x), \mathcal{O}(\text{nn}(x)), \delta)$      1: **function** $\mathcal{G}_\tau(s, \hat{t}, \hat{\gamma}, \delta, \epsilon)$
2:     $\hat{t}, \hat{\gamma} \leftarrow \arg\min_{t,\gamma} \text{LogisticLoss}(t, \gamma, \mathcal{O})$      2:     $t', \gamma' \leftarrow \phi^{-1}(\hat{t}, \hat{\gamma}, 1 - \epsilon)$
3:                $\triangleright$ i.e solve Eq 10      3:     $\alpha \leftarrow (1 - \epsilon) \, \mathcal{L}(x, t', \gamma')$
4:     $\tau \leftarrow \min_{\epsilon \in [0,1]} \mathcal{G}_\tau(x, \hat{t}, \delta, \epsilon)$      4:     $\tau \leftarrow \dfrac{(1 - \delta) - \alpha}{1 - \alpha}$
5:     $u \sim \text{Uniform}(0, 1)$     
6:     **if** $u \leq \tau$ **then**      5:     **return** $\tau$
7:        **return** explore      6: **end function**
8:     **else**
9:        **return** exploit
10:    **end if**
11: **end function**

---

all labeled examples observed for $y$. It then computes the $\hat{\tau}$ using Eq 11 by iterating over different values of confidence $\epsilon$. Then we use a uniform random variable $u \sim Uniform[0, 1]$ and explore if $u \leq \tau$ and exploit otherwise.

vCache makes two assumptions. First, the data $\mathcal{D}$ received by the cache is independently and identically drawn from the underlying distribution. Second, the true probability of correctness of response match given similarity, i.e. $\mathbf{Pr}(c(x) = 1 | \mathcal{D}, x)$ is well represented by the sigmoid family of functions (Eq 9).. Under these assumptions, the vCache policy can provide user-defined error-rate guarantees, as summarized in the following theorem.

**Theorem 4.1.** *Let $\delta \in (0, 1)$ be the user-provided maximum error tolerance. Let $\mathcal{D}, |\mathcal{D}| = n$ be the set of prompts seen by vCache at an arbitrary point in time. Then under the assumptions that prompts $\mathcal{D}$ are drawn i.i.d. from underlying distribution and sigmoid family of functions (defined in Eq 9) correctly model the true likelihood of correctness for each embedding, the probability of correct response from vCache for any arbitrary prompt $x$, executed in an online manner in accordance with Algorithm 2, is guaranteed to be greator than $1 - \delta$. i.e.*

$$\mathbf{Pr}(\mathbf{vCache}(x) = r(x) | \mathcal{D}) \geq (1 - \delta) \forall x, n \tag{12}$$

## 5   EVALUATION

For our experiments, we use three popular embedding models (GteLargeENv1-5 (Zhang et al., 2024), E5-large-v2 (Wang et al., 2022), and OpenAI text-embedding-3-small (ope, 2025)), and two LLMs (Llama-3-8B-Instruct (Dubey et al., 2024) and GPT-4o-mini (gpt, 2024)), representing both high-quality proprietary models and efficient open alternatives. We use the HNSW vector database (Malkov & Yashunin, 2018) with cosine similarity, a standard metric for comparing vector embeddings in semantic caching systems (Bang, 2023; Li et al., 2024; Dasgupta et al., 2025). All experiments are conducted on a machine running Ubuntu 24.04.2 LTS, with an Intel Xeon Platinum 8570 CPU and an NVIDIA Blackwell with 192 GB of memory.

**Metrics.** We measure the following metrics. (1) *Error Rate:* (Lower is better) defined as $FP/n$, where FP is the number of false positives and $n$ is the total number of prompts. (2) *Cache hit rate:* (Higher is better) defined as $(TP + FP)/n$ where TP and FP are true positives and false positives, respectively. Together, TP and FP measure the total cache hits. We also show ROC curves Hoo et al. (2017). For the non-deterministic evaluation of vCache (Algorithm 2, Line 5), we compute 95% confidence intervals using Wallis binomial confidence bounds and contingency tests Wallis (2013).

**Baselines.** We use the following Cache-settings in our experiments

- **GPTCache** (zilliztech): SOTA using a static threshold for all embeddings (Parameter: threshold).
- **GPTCache + Fine-tuned embedding**: Changing the embedding model in GPTCache. Embedding models are fine-tuned on data and method provided by Zhu et al. (2024) (Parameter: threshold).
- **vCache**: This is our proposed method (Parameter: error-rate bound $\delta$).
- **vCache + Fine-tuned embedding** (Zhu et al., 2024): Same as vCache, but uses a fine-tuned embedding model (Parameter: error-rate bound $\delta$).

We implement additional baselines for ablations and show the results in Appendix F.8.

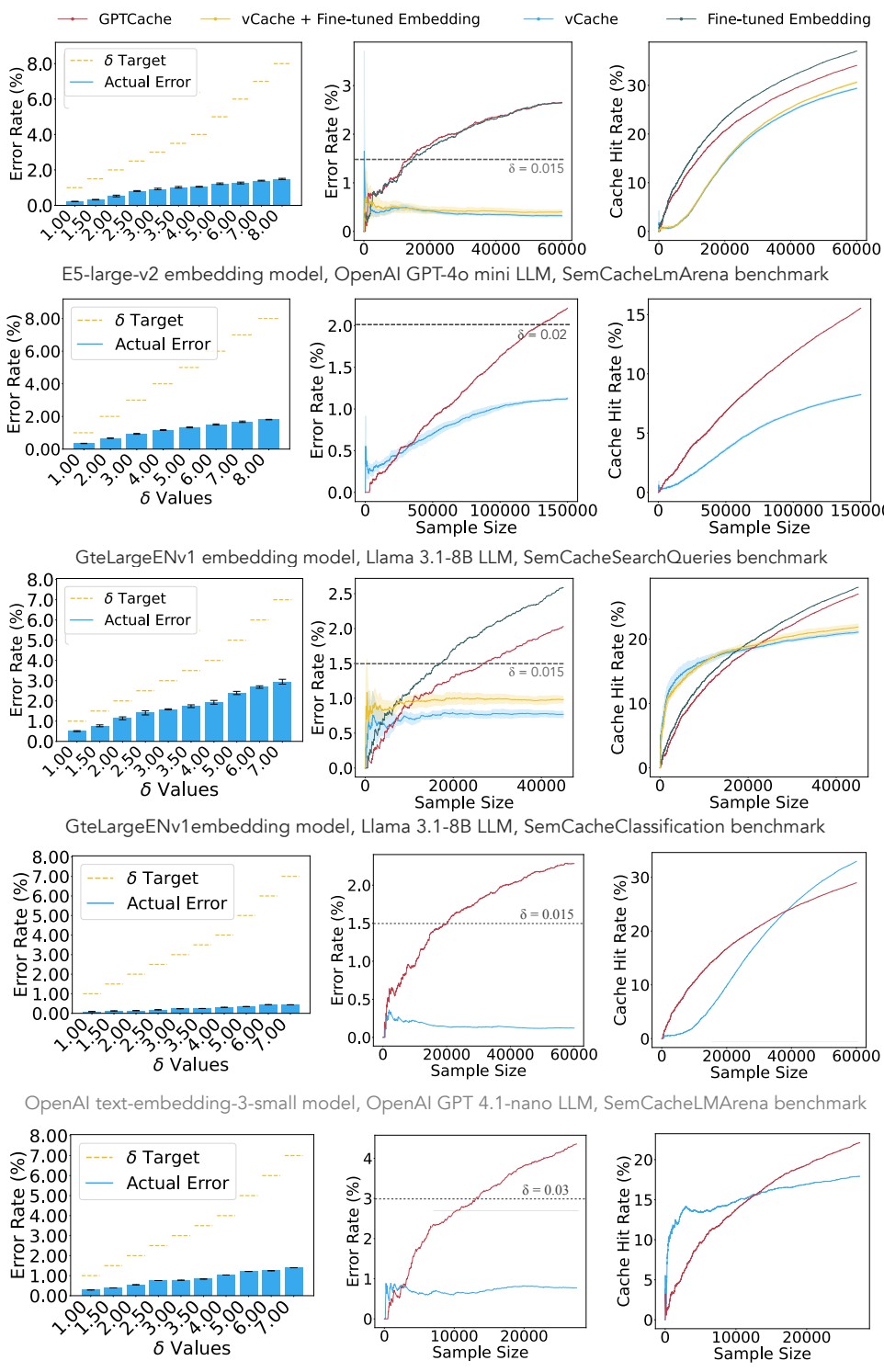

Figure 4: vCache meets the user-defined maximum error rate bound $\delta$ with steadily increasing cache hit rates (vCache is learning). GPTCache shows increasing error and hit rates, illustrating the unreliability of static thresholds. Static baselines use fixed thresholds. See Figure 5 for a threshold vs. $\delta$ Pareto comparison.

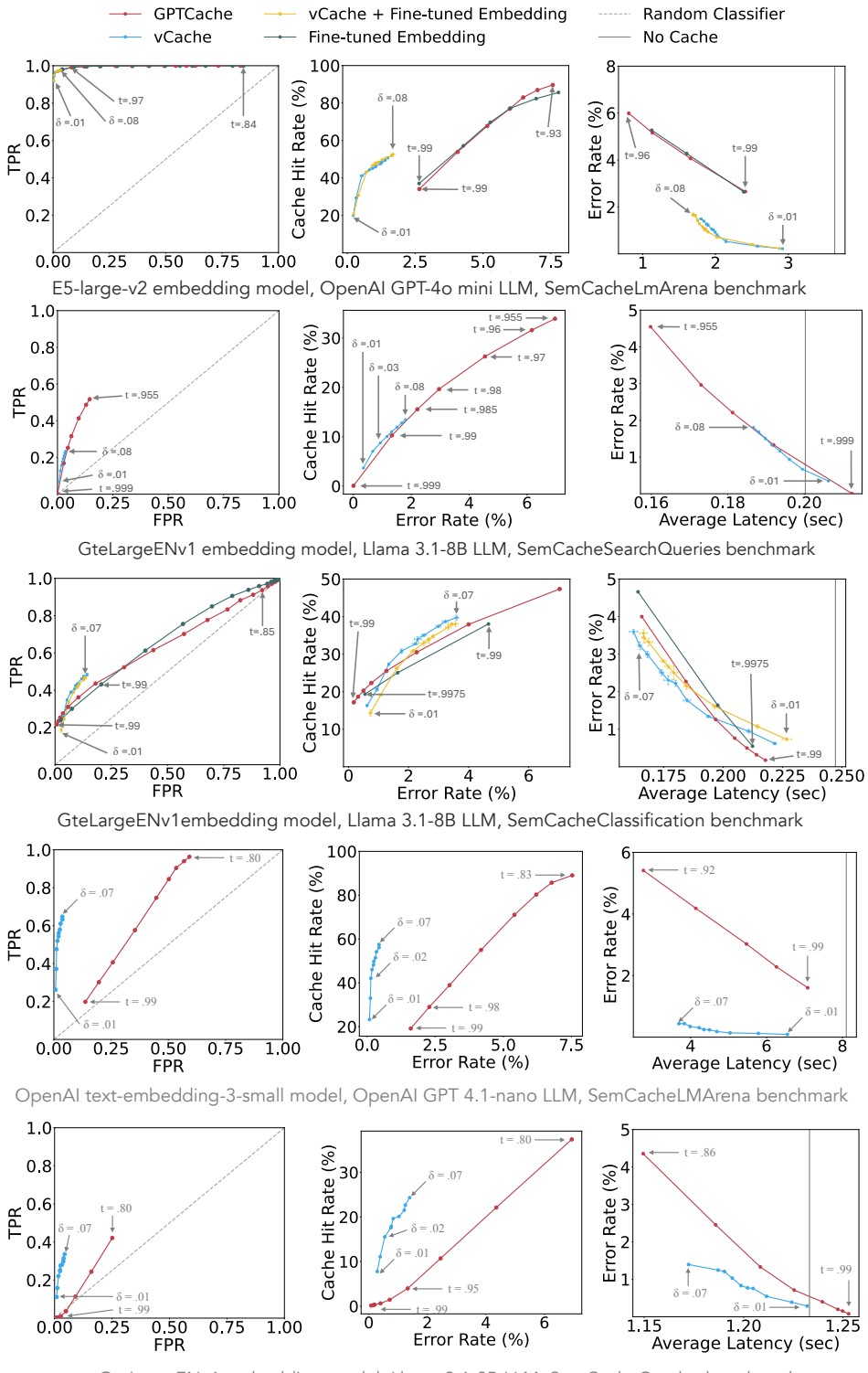

Figure 5: Pareto comparison across a range of thresholds and $\delta$ values. Each point represents a full run on all samples of a dataset.

**Datasets.** To the best of our knowledge, no realistic open-source benchmark currently exists for evaluating semantic caches. We introduce and open-source four diverse benchmarks designed to reflect common caching scenarios. Appendix H provides the complete dataset and benchmark cards.

- **SemCacheLMArena:** Randomly sampled subset of 60,000 queries from the LM-Arena human preference dataset (Chiang et al., 2024), containing open-ended, and user-generated prompts.
- **SemCacheClassification:** Benchmark of 45,000 prompts derived from three classification datasets (Saurabh Shahane, 2023; Talmor et al., 2018; Ni et al., 2019).
- **SemCacheSearchQueries:** Random subset of 150,000 web-search queries (Craswell et al., 2020).
- **SemCacheCombo:** 27,500-prompt benchmark combining SemCacheSearchQueries and distinct SemCacheLMArena queries to model partial workloads with no cache hits.

**vCache respects user-defined error-rate requirements.** We evaluate whether vCache satisfies the user-defined error rate bound $\delta$ while maintaining competitive performance. As shown in Figure 4, vCache consistently meets the maximum error rate across $\delta$ values, with actual error remaining below the specified bound. The small gap between maximum error rate and observed error stems from the conservative $t'$ estimation, which ensures robustness and can be further refined. Notably, as the error rate stabilizes, vCache continues to increase its cache hit rate, demonstrating effective learning over time. In contrast, GPTCache baselines exhibit increasing error rates with sample size, reflecting the inherent limitations of fixed thresholds despite improving hit rates.

**Dynamic and embedding-specific thresholds are superior to static thresholds.** We evaluate whether semantic caches benefit from dynamic, embedding-specific thresholds over a single static threshold. To this end, we compare vCache with static-threshold baselines by systematically varying threshold values for GPTCache and the maximum error rate ($\delta$) for vCache across a feasible range. Each point in Figure 5 represents a complete evaluation over the benchmark dataset, enabling direct Pareto comparison between vCache and static-threshold configurations. vCache achieves better ROC curves, higher cache-hit rates at a given error rate, and lower average latency. On SemCacheLMArena, it achieves up to 26× lower error, 12.5× higher hit rates, and reduced latency. On SemCacheClassification, vCache outperforms all baselines for error bounds above 1.5%, where static methods either violate constraints or underutilize the cache. For bounds below 1.5%, vCache is more conservative, reflecting its strategy of prioritizing correctness by increasing exploration under uncertainty. We note that GPTCache error rates have not fully converged and exhibit an upward trend (Figure 4), suggesting the reported results likely overstate GPTCache performance.

**Convergence, Sigmoid, Latency, vLLM, and Additional Baselines.** We report additional experiments on the convergence speed (Appendix D), aggregate empirical correctness probability (Appendix E), $\tau$ computation (Appendix F.4), embedding generation (Appendix F.5), logistic regression (Appendix F.6), vLLM inference (Appendix F.7), and extended baseline evaluations (Appendix F.8).

# 6    LIMITATIONS

There are two main limitations of vCache. First, for responses longer than a few words, string matching is insufficient and vCache uses LLM-as-a-judge (Zheng et al., 2023a) to assess response equivalence (Algorithm 1, L8), requiring an additional LLM inference. However, this can be executed asynchronously outside the critical path, not impacting latency (see vCache implementation). Since the output is a single token (e.g., "yes" or "no"), the overhead is minimal (Leviathan et al., 2023). In SemCacheLMArena and SemCacheSearchQueries, response equivalence is assessed with LLM-as-a-judge, whereas in SemCacheClassification it is determined by string matching. vCache performs reliably under evaluation regimes (Figure 4). Second, vCache relies on i.i.d. data and a sigmoid function family to represent the probability of correctness. If these assumptions are violated, the analysis may not hold. Nonetheless, both assumptions are natural and appear to model most use cases well, as supported by our experimental results (Appendix E).

# 7    CONCLUSION

We introduced vCache, a semantic cache that provides correctness guarantees by learning an optimal similarity threshold for each cached embedding online. This approach addresses the limitations of static thresholds and embedding fine-tuning in the semantic caching domain, ensuring that the error rate remains below a user-specified bound. Our experiments demonstrate that vCache consistently satisfies this guarantee while outperforming existing methods w.r.t error and cache hit rates. These results suggest that reliable, interpretable caching for LLMs is both practical and deployable.

## 8 ACKNOWLEDGMENT

The authors thank Sebastien Gautier-Jean-Jacques Beurnier and Siavash Ameli for their invaluable contributions. This research is partly supported by gifts from Accenture, Amazon, AMD, Anyscale, Broadcom, Google, IBM, Intel, Intesa Sanpaolo, Lambda, Lightspeed, Mibura, NVIDIA, Samsung SDS, and SAP.

## 9 ETHICS STATEMENT

This work contributes to the advancement of machine learning by improving the efficiency of large language model (LLM) inference. By reducing both computational cost and latency, our approach makes LLM-based systems more accessible to organizations and individuals with limited computational resources, thereby lowering the barrier to adoption. In addition, by decreasing the frequency of full LLM invocations, our method reduces overall compute demand and, consequently, the environmental impact associated with training and operating large-scale AI infrastructure.

## 10 REPRODUCIBILITY STATEMENT

We have taken several steps to ensure reproducibility of our work. The full implementation of vCache, including all algorithms and experiments, is publicly available in a GitHub repository[3]. The four semantic caching benchmarks introduced in this paper are released on HuggingFace[4], and Appendix H describes their construction and preprocessing in detail. All experimental settings, including hardware, software, and model configurations, are specified in Section 5. Theoretical assumptions and complete proofs of our correctness guarantees are provided in Appendix C. Together, these resources ensure that our results can be verified and extended by future work.

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

# A  Notation Glossary

For clarity, we summarize the key symbols and functions used throughout the paper in Table 1. The glossary covers cache contents, decision policies, probability functions, and modeling parameters. Each entry is accompanied by a short explanation and, where applicable, references to the defining equations, algorithms, or sections.

Table 1: A glossary of notations used in the paper and their explanations

| Notation | Explanation |
| --- | --- |
| $r(x)$ | Response produced by the LLM for prompt $x$. |
| $\mathcal{E}(x)$ | Vector embedding of prompt $x$. |
| $\mathrm{nn}(x)$ | Current nearest neighbor of $x$ in the cache; may change as the cache grows (see Equation 2). |
| $s(x)$ | Similarity between prompt $x$ and $\mathrm{nn}(x)$ (see Equation 4). |
| $c(x)$ | Correctness: 1 if $r(\mathrm{nn}(x)) = r(x)$, else 0 (see Equation 4). |
| $\mathcal{D}$ | Cache contents: $(\mathcal{E}(x_i), r(x_i), \mathcal{O}(x_i))_{i=1}^n$ (see Equation 3). |
| $\mathcal{O}(x_i)$ | Metadata for cached prompt $x_i$: all observed $(s(x_j), c(x_j))$ with $\mathrm{nn}(x_j) = x_i$ (see Equation 3). |
| $\mathcal{P}_{\mathrm{gptCache}}(s(x))$ | Static-threshold policy used by existing systems. |
| $\mathcal{P}_{\mathrm{vCache}}(s(x), \mathcal{O}(\mathrm{nn}(x)), \delta)$ | vCache policy using embedding-specific modeling under a user-defined error bound $\delta$ (Section 4). |
| $\delta$ | User-defined maximum error tolerance (see Section 4.1). |
| $\Pr(\mathrm{vCache}(x) = r(x))$ | Probability that vCache returns the correct response (see Equation 8). |
| $\tau_{\mathrm{nn}(x)}(s(x))$ | Required exploration probability to satisfy the guarantee (see Equation 8). |
| $\hat{\tau}$ | Upper bound used for exploration probability (see Equation 11). |
| $u \sim \mathrm{Uniform}(0, 1)$ | Random draw used to realize the exploration decision (see Algorithm 2). |
| $\mathcal{L}(s(x), t, \gamma)$ | Sigmoid likelihood modeling $\mathbf{Pr}(c(x) = 1 \mid x, \mathcal{D})$ (see Equation 9). |
| $t, \gamma$ | True threshold ($t$) and slope ($\gamma$) parameters of $\mathcal{L}$ (see Equation 9). |
| $\hat{t}, \hat{\gamma}$ | MLE estimates of $t$ and $\gamma$ based on $\mathcal{O}(\mathrm{nn}(x))$ (see Equation 10). |
| $t', \gamma'$ | Conservative estimates of the logistic parameters $t, \gamma$, selected from the confidence band of the MLE estimates. These values are used to compute $\hat{\tau}$ and ensure that vCache respects the user-defined error bound $\delta$ (see Equation 11). |

# B vCACHE SEMANTIC CACHE ARCHITECTURE

Figure 6 illustrates the vCache architecture. When a new prompt gets processed, it is first embedded into a vector representation and queried against the vector database to retrieve the nearest cached prompt. The similarity score and metadata of the retrieved embedding are passed to the similarity evaluator, which compares the correctness estimate against the user-defined error bound $\delta$ (see vCache decision policy in Section 4.2). If the policy returns exploit, the cached response is retrieved from the response store and served immediately. If the policy returns explore, the system performs an LLM inference to generate the true response, determines the correctness of the cached response with respect to the newly generated one, updates the metadata of the nearest neighbor, adds the new embedding to the vector database, adds the generated response to the response store, and returns it to the user (see Algorithm 1). In the vector database, the green balls represent the confidence bounds specific to the currently processed prompt. Larger balls indicate lower thresholds (more conservative exploitation). In comparison, smaller balls correspond to higher thresholds (more conservative exploration).

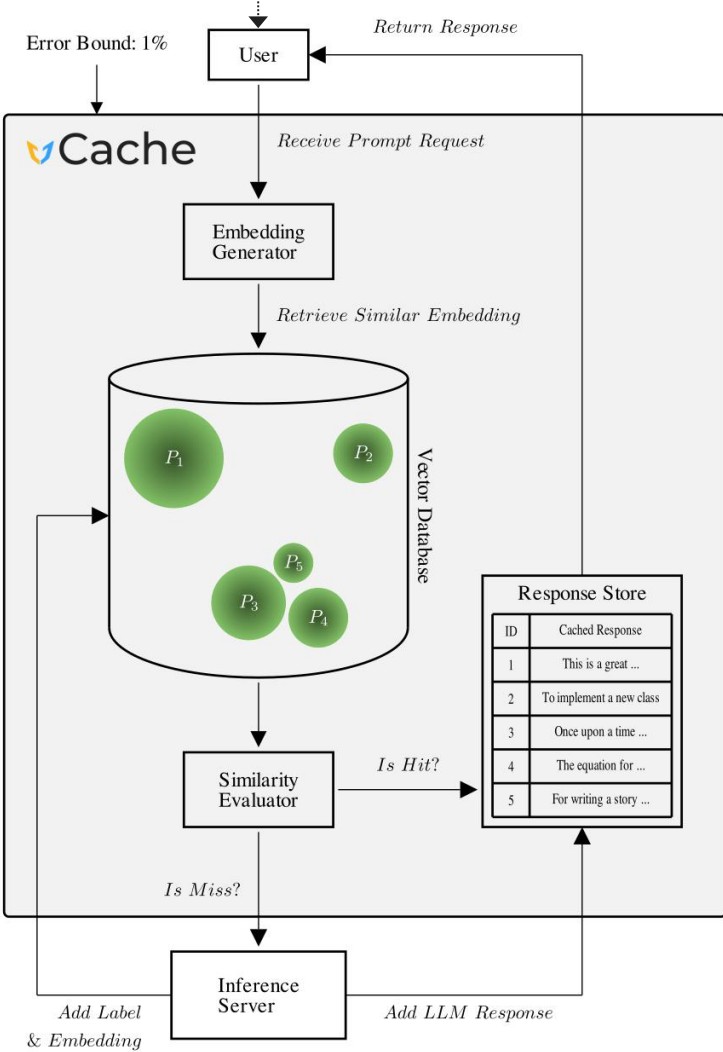

Figure 6: Workflow of the vCache architecture. Prompts are embedded, nearest neighbors retrieved, and the decision policy selects between exploiting a cached response or exploring via an LLM inference while ensuring the user-defined error bound $\delta$.

## C    VCACHE MODELING PROOF

We provide the proof for Theorem 4.1 in this section.

Recall we use the following notation in the paper,

- $x$ : prompt under consideration
- $\mathcal{D}$ : data inserted into the cache
- $\tau_{\text{nn}(x)}(s(x))$ : minimum probability of exploration associated with embedding $\text{nn}(x)$ at similarity value $s(x)$
- $\mathbf{Pr}_c = 1 - \delta$

Computing the exact probability $\mathbf{Pr}(c(x) = 1 \mid \mathcal{D}, x)$ is expensive, so we derive a simpler upper bound. If we have an upper bound, then computing an upper bound for $\tau_{\text{nn}(x)}(s(x))$ is straightforward.

**Lemma C.1** (Upper-Bounding $\tau$). *if* $\mathbf{Pr}(c(x) = 1|\mathcal{D}, x) \geq \alpha$, *then,*

$$\tau_{\text{nn}(x)}(s(x)) \leq 1 - \frac{\delta}{1 - \alpha}$$

*Proof.* Rewrite $\tau$ to isolate the unknown probability:

$$
\begin{aligned}
\tau_{\text{nn}(x)}(s(x)) &= \frac{(1 - \delta) - \mathbf{Pr}(c(x){=}1|\mathcal{D}, x)}{1 - \mathbf{Pr}(c(x){=}1|\mathcal{D}, x)} \\
&= 1 - \frac{\delta}{1 - \mathbf{Pr}(c(x){=}1|\mathcal{D}, x)}
\end{aligned}
\tag{13}
$$

Next, suppose we have a known lower bound

$$\mathbf{Pr}\big(c(x) = 1 \mid \mathcal{D}, x\big) \geq \alpha.$$

Then

$$1 - \mathbf{Pr}\big(c(x) = 1 \mid \mathcal{D}, x\big) \leq 1 - \alpha,$$

and since $\delta > 0$, it follows that

$$\frac{\delta}{1 - \mathbf{Pr}\big(c(x) = 1 \mid \mathcal{D}, x\big)} \geq \frac{\delta}{1 - \alpha}. \tag{14}$$

$\square$

Hence, to guarantee the exploration probability meets $\tau$, it suffices to ensure

$$\mathbf{Pr_{explore}}(x, \mathcal{D}_n) \geq \tau' = 1 - \frac{\delta}{1 - \alpha} \geq \tau_{\text{nn}(x)}(s(x)). \tag{15}$$

**Lemma C.2** (Lower-Bounding Cache-Correctness Probability). *Given $\mathcal{D}$, let $\hat{t}_{\text{nn}(x)}$ and $\hat{\gamma}_{\text{nn}(x)}$ be the MLE estimates computed as,*

$$\hat{t}_{\text{nn}(x)}, \hat{\gamma}_{\text{nn}(x)} = \arg\min_{t,\gamma} \mathbb{E}_{(s,c)\in\mathcal{O}_{nn(x)}} \left[ \Big( c \cdot \log(\mathcal{L}(s, t, \gamma)) \Big) + \Big( (1{-}c) \cdot \log(1 - \mathcal{L}(s, t, \gamma)) \Big) \right] \tag{16}$$

*Let $t^*$ and $\gamma^*$ be the true parameters such that $\mathcal{L}(s(x), t^*, \gamma^*)$ is the true probability of correct cache hits. Consider an arbitrary $\epsilon \in (0, 1)$. Let $t', \gamma'$ be such that,*

$$\mathbf{Pr}(t^* > t' || \gamma^* < \gamma') < \epsilon \tag{17}$$

*Then,*

$$\mathbf{Pr}(c(x) = 1|\mathcal{D}, x) \geq (1 - \epsilon)\mathcal{L}(s(x), t', \gamma') \tag{18}$$

*Proof.* By the law of total probability:

$$
\begin{aligned}
&\mathbf{Pr}(c(x) = 1 | \mathcal{D}, x) \\
&= \mathbf{Pr}(t^* > t' || \gamma^* < \gamma') \cdot \mathbf{Pr}(c(x) = 1 | \mathcal{D}, x, (t^* > t' || \gamma^* < \gamma')) \\
&\quad + \mathbf{Pr}(\neg(t^* > t' || \gamma^* < \gamma')) \cdot \mathbf{Pr}(c(x) = 1 | \mathcal{D}, x, t^* \leq t', \gamma^* \geq \gamma') \\
&\geq \mathbf{Pr}(\neg(t^* > t' || \gamma^* < \gamma')) \cdot \mathbf{Pr}(c(x) = 1 | \mathcal{D}, x, t^* \leq t', \gamma^* \geq \gamma') \\
&\geq (1 - \epsilon) \cdot \mathbf{Pr}(c(x) = 1 | \mathcal{D}, x, t^* \leq t', \gamma^* \geq \gamma').
\end{aligned}
\tag{19}
$$

$$
\begin{aligned}
&(1 - \epsilon) \cdot \mathbf{Pr}(c(x) = 1 | \mathcal{D}, x, t^* \leq t', \gamma^* \geq \gamma'), \\
&\geq (1 - \epsilon) \int_0^{t'} \int_{\gamma'}^{\infty} \mathbf{Pr}(t^* = t, \gamma^* = \gamma | t \leq t', \gamma \geq \gamma') \ \mathbf{Pr}(c(x) = 1 | \mathcal{D}, x, t^* = t, \gamma^* = \gamma) dt d\gamma \\
&\geq (1 - \epsilon) \left( \int_0^{t'} \int_{\gamma'}^{\infty} \mathbf{Pr}(t^* = t, \gamma^* = \gamma | t \leq t', \gamma \geq \gamma') dt d\gamma \right) \cdot \inf_{t \leq t', \gamma \geq \gamma'} \mathbf{Pr}(c(x) = 1 | \mathcal{D}, x, t) \\
&= (1 - \epsilon) \inf_{t \leq t', \gamma \geq \gamma'} \mathbf{Pr}(c(x) = 1 | \mathcal{D}, x, t) \\
&= (1 - \epsilon) \inf_{t \leq t', \gamma \geq \gamma'} \mathcal{L}(x, t, \gamma) \\
&= (1 - \epsilon) \cdot \mathcal{L}(x, t', \gamma')
\end{aligned}
\tag{20}
$$

since, $\mathcal{L}(x, t_1, \gamma) < \mathcal{L}(x, t_2, \gamma), \forall x. \ t_1 > t_2$ and $\mathcal{L}(x, t, \gamma_1) < \mathcal{L}(x, t, \gamma_2), \forall x. \ \gamma_1 < \gamma_2$  $\square$

Combining these results lets us set

$$
\alpha = (1 - \epsilon) \cdot \mathcal{L}(x, t', \gamma') \quad \implies \quad \tau'(\epsilon) = 1 - \frac{\delta}{1 - \alpha},
\tag{21}
$$

and then use $\tau'$. To find the best $\tau'$ closest to the actual lower bound of exploration probability, we search for the minimum $\tau'$ over the entire range of $\epsilon \in (0, 1)$ Thus,

$$
\tau' = \min_{\epsilon \in (0,1)} \left[ 1 - \frac{\delta}{1 - (1 - \epsilon) \mathcal{L}(s(x), t'(\epsilon), \gamma'(\epsilon))} \right]
\tag{22}
$$

**Confidence Bound on Optimal Threshold** To find $t', \gamma'$ from the estimated $\hat{t}, \hat{\gamma}$, we can use the confidence intervals by assuming a uniform prior on $t^*$ and $\gamma^*$. Since, under uniform prior the distributions of $\mathbf{Pr}(t^* | \hat{t})$ and $\mathbf{Pr}(\hat{t}, t^*)$ are the same,

$$
\begin{aligned}
\mathbf{Pr}(t^* | \hat{t}) &= \frac{\mathbf{Pr}(\hat{t} | t^*) \mathbf{Pr}(t^*)}{\mathbf{Pr}(\hat{t})} \\
\mathbf{Pr}(t^* | \hat{t}) &\propto \mathbf{Pr}(\hat{t} | t^*) \\
\mathbf{Pr}(t^* | \hat{t}) &= \mathbf{Pr}(\hat{t} | t^*)
\end{aligned}
\tag{23}
$$

Thus we can obtain $t', \gamma'$ using CDF of $\mathbf{Pr}(\hat{t}, \gamma | t^*, \gamma^*)$

In our experiments, we only use confidence intervals for $t$, i.e., we use the $t'$ parameter to adjust the likelihood. We estimate and use the point estimate $\hat{\gamma}$ for $\gamma$.

## D  CONVERGENCE SPEED ESTIMATION

For each cached prompt $y = \mathrm{nn}(x)$, vCache fits the sigmoid model $\mathcal{L}(s(x), t, \gamma)$ to the meta-data $\mathcal{O}(y)$. Under the standard regularity assumptions for logistic regression, the MLE $(\hat{t}_y, \hat{\gamma}_y)$ converges

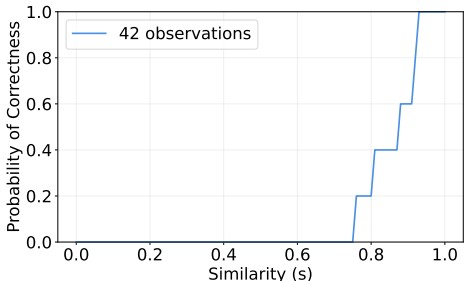 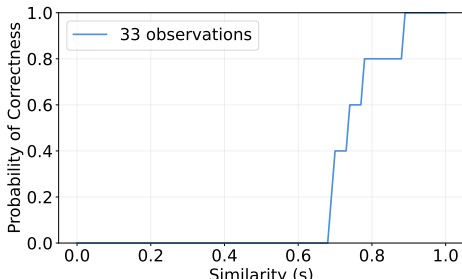

Figure 7: Empirical probability of correctness as a function of similarity $s$ for the two cached embeddings with the largest numbers of labeled observations (42 and 33) in SemBenchmarkLmArena with text-embedding-3-small and GPT-4.1-nano. We estimate $\Pr(c = 1 \mid s, nn(x))$ using a 1D $k$-NN smoother with $k = 5$. In both cases the curve is monotone and approximately sigmoidal in $s$, supporting the per-embedding sigmoid model $\mathcal{L}(s(x), t, \gamma)$ in Eq. 9.

to the true embedding-specific parameters $(t_y, \gamma_y)$ at the usual parametric rate $\mathcal{O}(1/\sqrt{n_y})$, where $n_y$ denotes the number of explored queries whose nearest neighbor is $y$. Any embedding-specific threshold derived from this model (e.g., the similarity at which $\Pr(c(x) = 1 \mid x, \mathcal{D})$ exceeds $1 - \delta$, or the required exploration probability $\tau_{\mathrm{nn}(x)}(s(x))$) is a smooth function of $(t_y, \gamma_y)$ and therefore inherits the same $\mathcal{O}(1/\sqrt{n_y})$ convergence rate via the delta method. Intuitively, as vCache collects more labels for a given neighbor, the decision boundary $t$ and the corresponding policy $P_{\mathrm{vCache}}(s(x), \mathcal{O}(\mathrm{nn}(x)), \delta)$ stabilize at this standard parametric speed.

# E    EMPIRICAL JUSTIFICATION OF THE SIGMOID MODEL

We model, for each cached embedding $nn(x)$, the probability $\Pr(c(x) = 1 \mid x, \mathcal{D})$ as a function of similarity $s(x)$ using the logistic family. We require the following structural properties from $\mathcal{L}(s(x), t, \gamma)$: 1) monotonicity in similarity, 2) boundedness in $[0, 1]$, and 3) a low-dimensional parameterization so that we can obtain tight confidence bands for $(t, \gamma)$ from the relatively small observation sets $\mathcal{O}_{nn(x)}$. The logistic sigmoid is a canonical choice satisfying these properties and allows efficient MLE.

To empirically validate this choice, we analyze the SemBenchmarkLmArena configuration with text-embedding-3-small and GPT-4.1-nano. We randomly select and evaluation run and for each cached embedding we collect all observed pairs $(s(x), c(x)) \in \mathcal{O}_{nn(x)}$, where $s(x)$ is the similarity between $x$ and $nn(x)$ and $c(x) \in \{0, 1\}$ indicates whether returning the cached response was correct. Figure 7 shows, for the two embeddings with the largest numbers of labeled observations (42 and 33, respectively), a 1D $k$-NN estimate ($k = 5$) of $\Pr(c = 1 \mid s, nn(x))$ as a function of similarity $s$. Both curves are monotone and exhibit a S-shaped transition from low to high correctness as similarity increases, which is the behavior that Eq. 9 is designed to capture for each $nn(x)$ and then feed into $\tau_{nn(x)}(s(x))$ and $\hat{\tau}$ in Eq. 9 and Eq. 11.

Figure 8 shows an aggregate view across all cached embeddings in the same configuration. We bin all $(s(x), c(x))$ pairs by similarity and plot the empirical correctness probability $\Pr(c = 1 \mid s)$ per bin. The resulting curve is again monotone and clearly S-shaped, with low correctness at small similarity, a rapid increase in a mid-range similarity region, and saturation near one at high similarity.

# F    ADDITIONAL ACCURACY, LATENCY, AND BASELINE EVALUATIONS

## F.1    SEMCACHESEARCHQUERIES BENCHMARK

We discuss the SemCacheSearchQueries benchmark, focusing on understanding the limitations of static-threshold caching. We highlight why fixed thresholds fail to maintain reliable error guarantees at scale and how vCache addresses this issue through dynamic, embedding-specific thresholding.

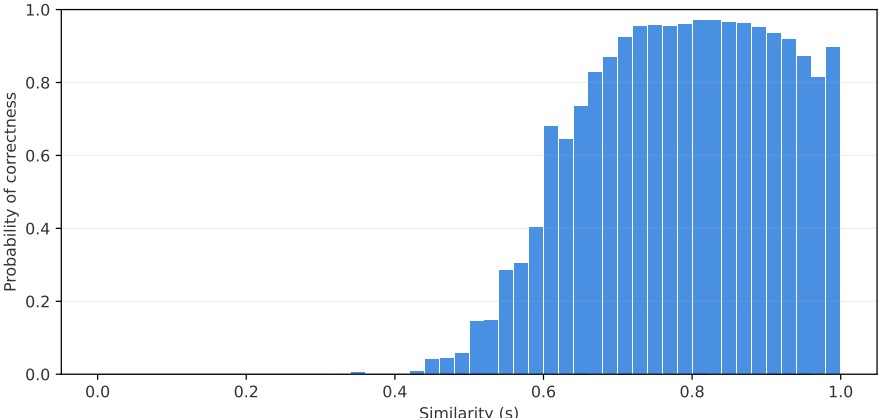

Figure 8: Aggregate empirical correctness probability as a function of similarity $s$ over all cached embeddings in SemBenchmarkLmArena with text-embedding-3-small and GPT-4.1-nano. We bin all $(s(x), c(x))$ pairs by similarity and plot the fraction of correct reuse decisions per bin. The resulting curve is monotone and clearly S-shaped, with low correctness at low similarity, a sharp increase in a mid-similarity region, and saturation at high similarity, which matches the qualitative behavior modeled by the sigmoid family in Eq. 9.

**vCache respects user-defined error-rate requirements**   We evaluate whether vCache satisfies the user-defined error rate $\delta$ while maintaining competitive performance. As shown in Figure 4, vCache consistently remains below the specified error bound across all tested $\delta$ values. Moreover, as the error rate stabilizes, vCache continues to improve its cache hit rate (Figure 4, right). In contrast, GPTCache exhibits increasing error rates as the sample size grows, despite improved hit rates. This trend reflects a fundamental limitation of static thresholds: maintaining a bounded error rate requires continuously increasing the threshold. Over time, no static threshold below 1.0 may suffice to satisfy a strict error constraint, making such systems difficult to tune and unreliable at scale.

**Dynamic and embedding-specific thresholds are superior to static thresholds**   We evaluate whether dynamic, embedding-specific thresholds yield better long-term performance than static thresholding. To this end, we compare vCache against GPTCache by varying static similarity thresholds and vCache's maximum error rate bound $\delta$. Each point in Figure 5 reflects a complete evaluation over the SemCacheSearchQueries benchmark, enabling a direct Pareto comparison. Static-threshold configurations achieve lower cache hit rates and higher error rates than vCache under equivalent evaluation settings. As shown in Figure 4, GPTCache has an increasing error rate as the sample size grows because the threshold remains fixed (both cache hit rate and error rate rise together). As a result, the GPTCache curve in the middle plot (cache hit vs. error rate) is expected to shift up and to the right, while the curve in the right plot (error rate vs. latency) shifts up and to the left. This trend suggests that no static threshold below 1.0 can maintain a bounded error rate as prompt diversity increases (see the threshold of 0.99, which yields an error rate of 1.7% after 150k samples). In contrast, vCache learns its threshold online and per embedding, allowing it to enforce the error constraint while gradually improving cache hit rate.

### F.2   SEMCACHECOMBO BENCHMARK

**vCache respects user-defined error-rate requirements on SemCacheCombo**   Figure 4 evaluates whether vCache satisfies the user-defined error rate $\delta$ on the SemCacheCombo benchmark. Across all tested $\delta$ values, the realized error of vCache remains below the requested bound, confirming that the learned thresholds reliably enforce the target error rate. As more samples are processed, vCache maintains its empirical error while steadily increasing cache hit rate, indicating ongoing learning from additional data. In contrast, GPTCache with a fixed similarity threshold of 0.83 exhibits growth in both error rate and hit rate as the sample size increases.

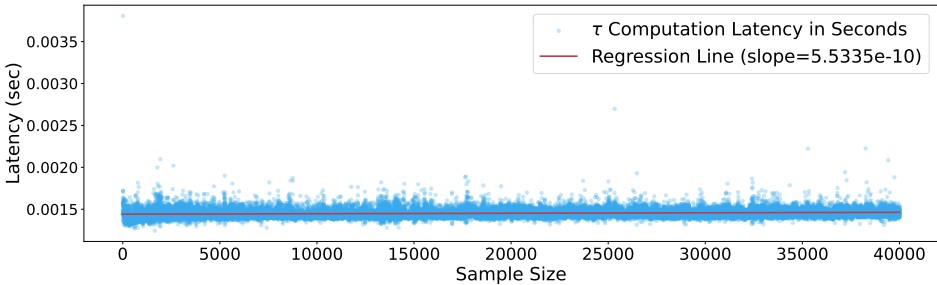

Figure 9: Empirical latency of computing $\tau$ in Algorithm 1 as a function of sample size. The fitted regression line (slope $\approx 5.5 \times 10^{-10}$) shows that the cost is effectively constant, adding at most $\approx 1.5$ ms ($< 0.0015$ s) per update.

**Dynamic, embedding-specific thresholds yield better trade-offs than static thresholds on Sem-CacheCombo** Figure 5 summarizes the resulting trade-offs by varying vCache's error bound $\delta$ and GPTCache's static similarity threshold. Each point corresponds to a full evaluation on Sem-CacheCombo, enabling a direct Pareto comparison. vCache traces a strictly better frontier: it achieves higher cache hit rates at the same error level, and lower error for comparable hit rates and latency. For example, vCache attains up to $12.5\times$ higher cache hit rates than the best static-threshold configuration while still satisfying the user-defined error bound. At matched average latency, vCache consistently delivers lower error than GPTCache. These results show that dynamic, embedding-specific thresholds provide superior accuracy–efficiency trade-offs to static thresholds.

### F.3 SemCacheLMArena with OpenAI Embeddings

**vCache respects user-defined error-rate requirements on SemCacheLMArena** Figure 4 evaluates whether vCache satisfies the user-defined error rate $\delta$ on the SemCacheLMArena benchmark. Across all tested $\delta$ values, the realized error rate of vCache remains below the user-defined target, confirming that the learned thresholds enforce the desired bound. As the sample size grows, vCache further reduces its empirical error while steadily increasing cache hit rate, indicating that it continues to learn from additional data. In contrast, GPTCache with a fixed similarity threshold of $0.98$ exhibits growth in both error rate and hit rate as more prompts arrive. Figure 5 outlines a Pareto comparison across all feasible thresholds.

**Dynamic, embedding-specific thresholds yield better trade-offs than static thresholds** Figure 5 summarizes the overall accuracy–efficiency trade-offs on SemCacheLMArena by varying vCache's error bound $\delta$ and GPTCache's static similarity threshold. Each point represents a complete evaluation run, enabling a direct Pareto comparison. vCache traces out a strictly better frontier: it achieves substantially higher cache hit rates at the same error level and much lower error for comparable hit rates and latency. For example, vCache reaches a cache hit rate of $57\%$ while keeping the error rate below $0.5\%$, whereas GPTCache requires an order-of-magnitude higher error to obtain similar hit rates. At matched latency, vCache achieves up to $26\times$ lower error than the best static configuration. These outcomes demonstrate that dynamic, embedding-specific thresholds are better suited than static thresholds for maintaining strict error guarantees while exploiting semantic redundancy.

### F.4 $\tau$ Computation Overhead

Figure 9 reports the latency of computing $\tau$ (Algorithm 1) on the SemCacheLMArena benchmark as a function of the current sample size. Each point corresponds to one update step, and the solid red line is a linear regression fit. The points form a tight horizontal band and the regression slope is effectively zero ($\approx 5.5 \times 10^{-10}$ sec per sample), indicating that $\tau$ can be computed in constant time. Across all sample sizes, the latency remains below $1.5$ ms, indicating that the overhead of computing $\tau$ is negligible.

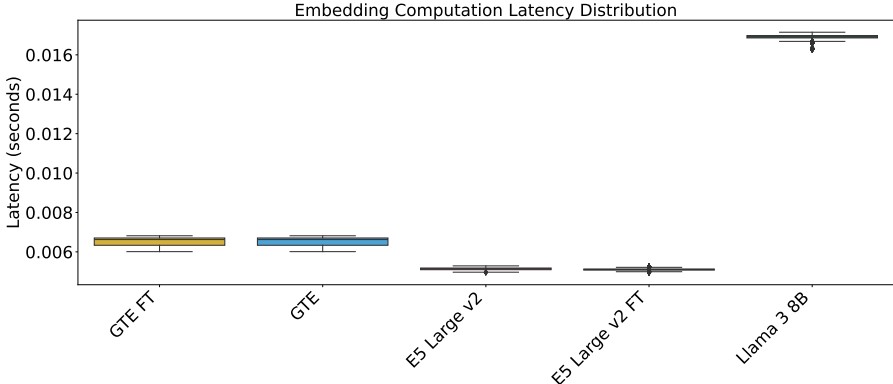

Figure 10: Embedding computation latency distributions across models, shown as 95th percentile whisker plots. GTE_FT = GteLargeENv1-5 (Zhang et al., 2024) fine-tuned (Zhu et al., 2024). E5 Large v2 FT = E5-large-v2 (Wang et al., 2022) fine-tuned (Zhu et al., 2024).

### F.5 EMBEDDING GENERATION OVERHEAD

Semantic caches incur additional latency due to embedding computation, which must be evaluated in relation to the cost of LLM inference. To quantify this overhead, we compare the embedding latencies of four models to the inference latency of `Llama3.1-8B`.

All experiments are conducted on a system with an Intel Xeon Platinum 8570 CPU and an NVIDIA Blackwell GPU with 192 GB of memory. The results show that embedding computation is significantly faster than model inference, confirming its applicability in caching pipelines.

### F.6 LOGISTIC REGRESSION LATENCY OVERHEAD

Since vCache performs threshold estimation online for every request, the efficiency of logistic regression directly impacts scalability. In our experiments, we use sklearn.linear_model on CPU, yielding an average latency of 0.0017 seconds on the SemBenchmarkArena. Its negligible latency ensures that online modeling does not introduce noticeable overhead and can scale to large caches.

### F.7 VCACHE IN COMBINATION WITH VLLM

vCache is orthogonal to inference optimization systems, as semantic prompt caching reuses responses for semantically similar prompts rather than accelerating inference itself. When a cache miss occurs, vCache directly benefits from systems such as vLLM (Kwon et al., 2023) or SGLang (Zheng et al., 2023b), which reduce model latency. To validate this, we hosted LLaMa 3.1 70B with vLLM on two NVIDIA Blackwell GPUs and compared inference latency with and without vCache. For vCache, we additionally ran the GteLargeENv1_5 embedding model on the same machine. Evaluation was conducted on 45k samples from the SemCacheClassification benchmark.

Table 2: A comparison of overall runtime, latency, cache hit rate, and error rate with and without vCache under different error tolerances

| Baseline | Config | Overall Duration | Avg. LLM Inference Latency | Avg. Emb. Latency | Cache Hit Rate | Error Rate |
|---|---|---|---|---|---|---|
| vLLM | – | 240 min | 0.32 sec | – | 0.0% | 0.0% |
| vLLM + vCache | $\delta = 0.01$ | 214 min | 0.32 sec | 0.018 sec | 18.1% | 0.4% |
| | $\delta = 0.02$ | 170 min | 0.32 sec | 0.018 sec | 35.5% | 1.2% |
| | $\delta = 0.03$ | 160 min | 0.32 sec | 0.018 sec | 40.2% | 1.4% |

Despite the additional embedding overhead, vCache substantially reduces end-to-end latency by avoiding repeated LLM inferences. This demonstrates that vCache is complementary to inference optimization systems such as vLLM.

## F.8 ADDITIONAL BASELINE EVALUATIONS

vCache is the first adaptive, probabilistic, and Bayesian method for semantic caching. The most competitive alternatives are static threshold methods, which we extend with several additional baselines for completeness. The landscape of approaches can be ordered from naive to advanced as follows:

**GS:** **G**lobal and **S**tatic threshold (i.e., GPTCache).

**GD:** **G**lobal and **D**ynamic threshold (vCache with global threshold).

**LS:** Per-embedding (**L**ocal) and **S**tatic threshold.

**LD:** Per-embedding (**L**ocal) and **D**ynamic threshold:

> **LD1:** Logistic regression to compute the threshold ($\hat{t}$).
> **LD2:** Logistic regression sigmoid fit to model correctness probability.
> **LD3:** Logistic regression sigmoid fit with confidence intervals and guarantees (vCache).

We implement all baselines and perform ablations on the `E5-large-v2` embedding model, OpenAI `GPT-4o mini` LLM, and the `SemCacheLmArena` benchmark. For non-adaptive baselines (LD1 and LD2), we select the threshold or $\delta$ values that produced error rates closest to their observed performance. Results are shown in Table 3.

Table 3: A comparison of error rate, cache hit rate, and qualitative observations across baselines

| Baseline | Threshold/ Delta | Error Rate | Cache hit Rate | Comments and Observations |
|---|---|---|---|---|
| GS | 0.99 | 2.5% | 37% | No guarantee and worst trade-off. |
|  | 0.98 | 4.1% | 53% |  |
|  | 0.97 | 5.2% | 67% |  |
| GD | 0.02 | 1.3% | 14% | Due to the large overlap between incorrect and correct samples at a given similarity (see 3), the optimal threshold converges to 1.0, yielding low cache hits. |
|  | 0.03 | 2.5% | 26% |  |
|  | 0.05 | 4.3% | 45% |  |
| LS | – | – | – | Impossible to compute a threshold for every embedding a priori. |
| LD1 | – | 2.6% | 70% | No guarantees and no error-rate fine-tuning. |
| LD2 | – | 2.1% | 68% | No guarantees and no error-rate fine-tuning. |
| LD3 | 0.02 | 0.5% | 41% | Guarantees and beats both SOTA and GD baselines. |
|  | 0.03 | 1.1% | 46% |  |
|  | 0.05 | 2.0% | 54% |  |

This ablation underlines that (1) semantic caches benefit from embedding-specific thresholds, and (2) probabilistic modeling is required to satisfy user-defined error bounds and ensure predictability. Notably, only vCache provides guarantees. The importance of guarantees cannot be overstated, as in practice, the absence of correctness guarantees has been a primary reason for the failure of semantic caching deployments in industry.

## F.9 THRESHOLD DILEMMA

To analyze the relationship between similarity scores $s(x)$ and cache correctness $c(x)$, we conduct an experiment on the 45,000 entries of the *SemCacheClassification* benchmark using an error tolerance of $\delta = 0.02$ (2%). For each cached embedding $x_i$, we record the observations $\mathcal{O}(x_i)$ and the empirically estimated optimal threshold $\hat{t}_{x_i}$.

In the top plot of Figure 3, we separate all observations into two sets: one where $c(x) = 1$ (correct cache hit) and another where $c(x) = 0$ (incorrect hit). We plot the kernel density functions (KDF), also known as kernel density estimations (KDE), to visualize the distribution of correct and incorrect observations. The result shows that correct and incorrect observations are nearly indistinguishable in similarity space, with substantial overlap and similar means (0.84 vs. 0.85). This illustrates that one single similarity threshold is not a reliable decision boundary.

The bottom plot shows a histogram of the optimal threshold values $\hat{t}$ computed per embedding. The thresholds span a range, from 0.71 to 1.0, indicating that no single similarity threshold can suffice across embeddings. A threshold set too low increases the risk of incorrect cache hits; a threshold set too high limits cache hits. Together, these results motivate the need for embedding-specific and dynamically learned thresholds to ensure interpretable and reliable performance.

## G HYPERPARAMETER GUIDANCE

vCache is designed to be simple to configure, with the error rate bound $\delta$ as its primary hyperparameter. We recommend setting this bound based on the desired trade-off between accuracy and cost. For high-accuracy applications (e.g., customer support or safety-critical systems), a conservative value such as 0.5% can be appropriate. For use cases with higher tolerance for occasional errors and stronger cost or latency constraints (e.g., search or summarization), values around 2–3% may be reasonable. Ultimately, the choice depends on application-specific requirements.

## H BENCHMARK CREATION

To the best of our knowledge, no open-source benchmarks currently exist for evaluating the performance and applicability of semantic caching systems. To address this gap, we construct and release four diverse benchmarks[5], each designed to reflect a distinct real-world use case: classification tasks, conversational chatbots, and search engines. This section describes the motivation and construction process behind each benchmark.

### H.1 SEMCACHECLASSIFICATION BENCHMARK

The *SemCacheClassification* benchmark is designed to evaluate semantic caching in structured classification settings, such as those found in modern database environments (Liu et al., 2024b). Several database systems, including Databricks, Snowflake, and AWS, have introduced LLM-based extensions to SQL via User-Defined Functions (UDFs), enabling capabilities beyond traditional SQL semantics (Liu et al., 2024b). However, LLM-based UDFs are inherently non-deterministic, execute row by row, and pose integration challenges for parallelization and query optimization (Franz et al., 2024). When table rows contain semantically similar values, semantic caching can reduce the frequency of LLM invocations, thereby improving both latency and cost. This benchmark captures such use cases, where slight variations in phrasing should not require repeated inference.

The benchmark consists of 45,000 short-form prompts with a fixed output label space. Each example follows a prompt–response format, where the prompt expresses a classification query and the expected response is a one-word label. The benchmark combines three diverse classification datasets: *CommonsenseQA* (Talmor et al., 2018), *Ecommerce Categorization* (Saurabh Shahane, 2023), and *Amazon Instant Video Review* (Ni et al., 2019). This dataset composition models out-of-distribution data because the three sources differ significantly in domain, style, and vocabulary, forcing semantic caching methods to generalize beyond a single homogeneous dataset. Sample prompts and response

---

[5]https://huggingface.co/vCache

formats from each dataset are shown below, and Table 4 summarizes label distributions across the benchmark.

A sample entry from the Ecommerce Categorization (Saurabh Shahane, 2023) dataset:

```
{
    "prompt": "Which category does the text belong to? Text: <text
    >",
    "output_format": "Answer with 'Books', 'Electronics', '
    Household', or 'Clothing & Accessories' only"
}
```

A sample entry from the Commonsense QA Talmor et al. (2018) dataset:

```
{
    "prompt": "What is the main subject of the following question?
    Question: <question>",
    "output_format": "Answer with only one of the words of this set
    : ['people', 'potato', 'competing', , 'snake', 'lizard', 'food'
    , 'car', 'water', 'student', 'crab', 'children', 'killing', '
    animals', 'ficus', 'horse', 'fox', 'cat', 'weasel', 'shark', '
    person', 'human']"
}
```

A sample entry from the Amazon Instant Video Review Ni et al. (2019) dataset:

```
{
    "prompt": "Is this review friendly? Review: <review>",
    "output_format": "Answer with 'yes' or 'no' only"
}
```

The benchmark enables controlled evaluation of semantic caching strategies, especially in cases where small changes in input phrasing must still map to the same output class. Its fixed label format makes it particularly useful for evaluating systems like vCache, which rely on correctness guarantees under threshold uncertainty. Table 4 summarizes the label distributions for the three subtasks in the SemCacheClassification benchmark.

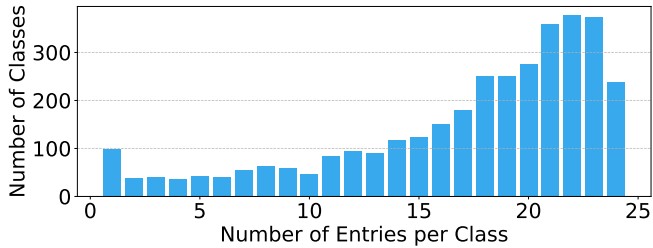

Figure 11: Distribution of class sizes in the SemCacheLMArena benchmark. Each class corresponds to a unique user prompt and contains 2–24 semantically similar variants.

| Response | Count |
|---|---|
| person | 2,806 |
| people | 625 |
| human | 400 |
| competing | 272 |
| animals | 225 |
| food | 202 |
| car | 125 |
| water | 100 |
| student | 79 |
| children | 32 |
| killing | 27 |
| horse | 24 |
| lizard | 13 |
| potato | 13 |
| fox | 11 |
| cat | 10 |
| ficus | 10 |
| weasel | 8 |
| shark | 8 |
| crab | 7 |
| snake | 3 |

(a) Commonsense QA.

| Response | Count |
|---|---|
| Books | 6,000 |
| Clothing | 6,000 |
| Electronics | 6,000 |
| Household | 2,000 |

(b) Ecommerce.

| Response | Count |
|---|---|
| yes | 10,000 |
| no | 10,000 |

(c) Amazon Instant Video.

Table 4: Response distribution across three datasets that form the SemCacheClassification benchmark.

## H.2 SEMCACHELMARENA BENCHMARK

The *SemCacheLMArena* benchmark is designed to evaluate semantic caching in chatbot environments, where users may issue semantically similar prompts with different phrasing. In such settings, caches must generalize across diverse surface forms while maintaining response correctness. This benchmark addresses these challenges by grouping semantically similar user inputs and testing whether caching systems can accurately reuse responses.

To construct the benchmark, we use the LM-Arena human preference dataset (Zheng et al., 2023a), which contains 100,000 real-world user queries. We randomly sample 3,500 distinct prompts, each of which defines a class. For each class, we generate between 1 and 23 semantically similar variants using GPT-4.1-nano, resulting in a total of 60,000 prompts. A class ID is assigned to each prompt to evaluate caching correctness: a cache hit is considered correct if the retrieved response belongs to the same class as the query. Figure 11 shows the distribution of class sizes (number of prompts belonging to a class), confirming broad variability in prompt paraphrasing. To support model-agnostic evaluation, we generate responses for all prompts using GPT-4.1-nano and GPT-4o-mini. The corresponding response length distributions are shown in Figure 12.

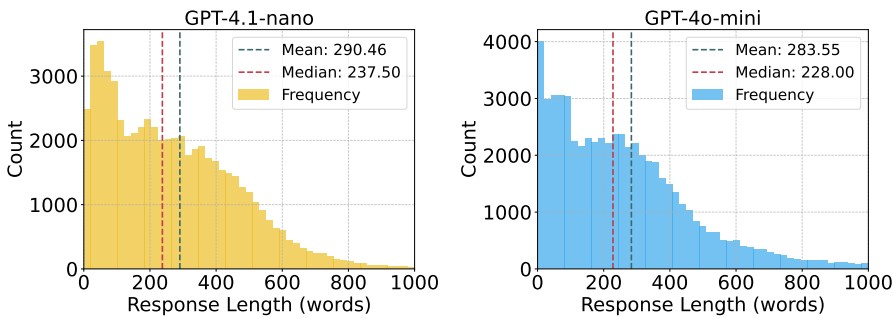

Figure 12: Response length histogram for GPT-4.1-nano and GPT-4o-mini on the SemCacheLMArena benchmark. Median and mean lengths are shown for each model.

## H.3 SemCacheSearchQueries Benchmark

The *SemCacheSearchQueries* benchmark is designed to evaluate semantic caching in open-domain search applications. Large-scale search engines, such as Google, increasingly rely on LLMs to generate direct answers to natural language queries (Liu et al., 2024a; Wang et al., 2024). While this improves user experience, it introduces significant latency and cost, particularly at the scale of millions of daily queries. Many queries issued to search engines are paraphrased variations of earlier inputs, making semantic caching a natural fit for reducing redundant LLM inference in this setting.

The benchmark is constructed from a filtered subset of the ORCAS dataset (Craswell et al., 2020), containing real-world search engine queries. We begin by sampling 500,000 queries and embedding each using the gte-large-en-v1.5 embedding model (Zhang et al., 2024). We then apply $k$-means clustering to group similar queries and retain the largest clusters, resulting in 150,000 entries. Within each cluster, we apply a union-find algorithm guided by an LLM-based judge (GPT-4.1-nano) to determine whether query pairs yield the same response. Sub-clusters identified in this step define the equivalence classes used for caching evaluation. Figure 13 summarizes the benchmark properties, including class size distribution, frequent query terms, and statistics on the number of queries per class.

## H.4 SemCacheCombo Benchmark

We introduce the SemCacheCombo benchmark to evaluate semantic caching in workloads that contain both reusable responses and unique, non-reusable responses. The dataset is constructed by combining two sources into a single sequence of 27,500 prompts. First, we take the SemCacheLMArena benchmark and select one representative prompt from each of its 3,500 semantic classes. Because each prompt represents a different cluster, these 3,500 queries are pairwise semantically distinct and should not share a reusable response. From the perspective of a semantic cache, every SemCacheLMArena prompt is therefore expected to result in a cache miss; any cache hit on this subset is, by definition, an incorrect reuse. Second, we append 24,000 prompts from the SemCacheClassification benchmark. Correct cache hits may only occur within this set of prompts. Next, we randomly shuffle all 27,500 prompts.

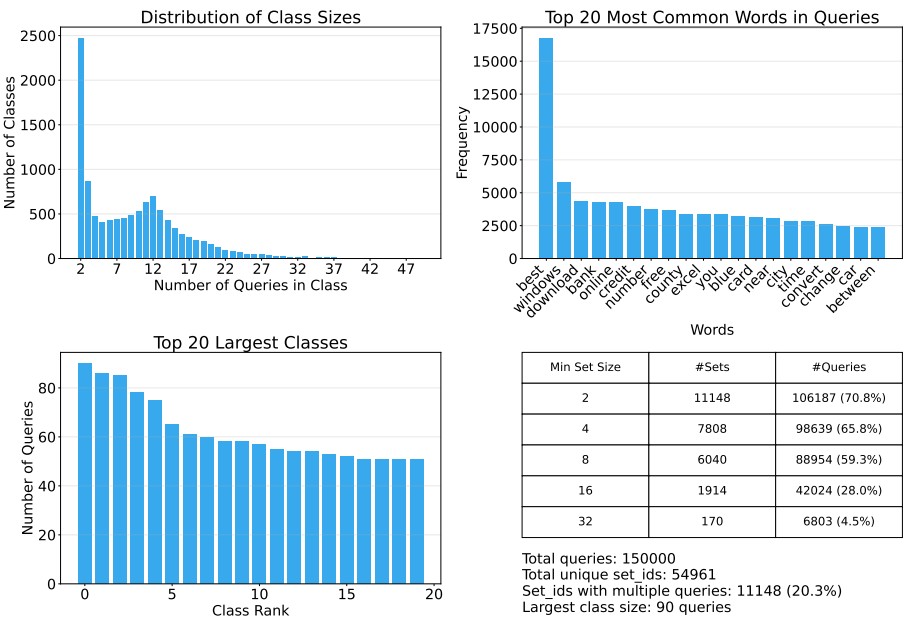

Figure 13: Descriptive statistics for the SemCacheSearchQueries benchmark. Top left: Most classes contain exactly two queries. Top right: The word *best* appears in over 16,000 of the 150,000 queries. Bottom left: The largest class contains 90 semantically equivalent queries. Bottom right: 59.3% of all classes contain more than eight queries, indicating substantial intra-class variability.

