# OpenReview forum: "vCache: Verified Semantic Prompt Caching"
_ICLR.cc/2026/Conference — ICLR 2026 Poster_

### Official Review · Reviewer_sFQ7 · 2025-10-27

**Soundness:** 2
**Presentation:** 3
**Contribution:** 2
**Rating:** 4
**Confidence:** 4

**Summary:**

This paper proposes using dynamic thresholds for different prompts in semantic caching, thereby providing an error rate guarantee. Based on a sigmoid parametric model and observed samples, the proposed online learning algorithm continually estimates the optimal thresholds for cached prompts. Experimental results demonstrate its effectiveness in controlling error rates.

**Strengths:**

- The paper is well structured and easy to follow.
- Guaranteed error rate is a crucial issue for semantic caching, and the proposed online-learning-based dynamic threshold method provides a reasonable solution.

**Weaknesses:**

- Although vCache can guarantee the error rate of the semantic cache, it heavily relies on the observation of correctness (Algorithm 1, Line 8). In the experimental section, the authors propose using exact matching for short prompts and LLM inference for long prompts to determine correctness. The additional LLM inference introduces extra cost for the semantic cache, which undermines its practical value. Moreover, the LLM could make mistakes in judgment, making the guarantee unreliable.
- Why is the actual error rate in Figure 4 much lower than $\delta$? According to the guarantee, they should not differ by such a large margin.
- Despite the controlled error rate, the risk of private data leakage still exists, which limits its practical adoption in industry.
- Since LMArena contains many testing prompts, the high hit rate shown in Figure 4 might result from these testing cases. Can the authors provide details about the hit prompts for the three benchmarks?

**Questions:**

See weaknesses

---

> ### Author Response · Authors · 2025-11-21
> **Response to sFQ7 (Part I)**
>
> We thank the reviewers for their careful reading and constructive feedback, and we address their main concerns point by point below.
>
> ## Reliance on correctness samples
> We agree that vCache relies on observing whether returning a cached response was correct (Algorithm 1, Line 8); this is fundamental to any method that aims to estimate and control an error rate. Without some form of correctness feedback, it is impossible to enforce a user-defined error-rate bound. vCache only assumes access to a generic binary correctness signal and is agnostic to how it is obtained: in our benchmarks, correctness comes from exact matching for short classification-style prompts and from an LLM-judge for long-form prompts. That said, this signal could also be derived from existing application logic, offline labels, human-in-the-loop feedback, etc.. Crucially, the logistic model is updated from a subset of queries for which correctness is observed; it does not require labels for every request (i.e., during an exploit).
>
> ## Additional inference cost
> The LLM-based correctness judge used in our long-form benchmarks is one conservative instantiation of this generic correctness oracle, not a requirement of the framework. For each checked pair of responses, the judge is prompted to output only a concise decision (e.g., a 1–2 token “yes/no”-style judgment), whereas the underlying LLM responses in SemCacheLMArena have median lengths of ≈230–240 words. Thus, the additional decoding cost of the judge is negligible compared to generating the original answer.
>
> Operationally, the correctness check is decoupled from the user-facing path: the user either receives a cached response or invokes the underlying LLM, and the only synchronous overhead is embedding (< 7 ms, Figure 14) and /tau computation (< 1.5 ms, Figure 13).
>
> ## LLM-as-a-judge
> We agree that the choice of the evaluation oracle is important. In our setting, the judge is not part of the vCache algorithm; it is only used to obtain binary correctness labels $c(x) \in {0,1}$ for long-form responses. Our theoretical guarantee and decision rule only assume access to such labels and are agnostic to whether they come from humans, heuristics, or an LLM.
>
> To mitigate dependence on a specific judge, we evaluate vCache under three regimes:
>
> 1. SemCacheClassification benchmark, where correctness is determined by exact string matching over a fixed label set;
> 2. SemCacheLMArena and SemCacheSearchQueries benchmarks, where equivalence of mid-form responses is assessed with a standard LLM-as-a-judge protocol; and
> 3. SemCacheCombo, which mixes both types of workloads (newly added).
>
> Across all regimes, vCache satisfies the user-defined error bound and outperforms state-of-the-art baselines.
>
> For open-ended prompts, exact matching is infeasible, and LLM-as-a-judge has become standard practice, with prior work (see “Judging LLM-as-a-Judge with MT-Bench and Chatbot Arena”) showing high agreement with human annotations.
>
> ## Quantify gap between error bound and actual performance
> We agree that the gap between the specified error rate bound $\delta$ and the realized error rate in Figure 4 is important to understand. vCache is designed as a one-sided, high-probability guarantee: we control the upper tail of the error curve by using an upper confidence band when choosing thresholds. As long as there is residual statistical uncertainty, this construction is intentionally conservative and will tend to produce realized error rates below $\delta$; this is expected behaviour for a method that aims to guarantee error ≤ $\delta$ uniformly over time and across many embeddings, rather than matching $\delta$ exactly on average.

---

> ### Author Response · Authors · 2025-11-21
> **Response to sFQ7 (Part II)**
>
> ## Privacy leakage
> We agree that privacy and potential data leakage are concerns in industrial LLM deployments. However, this paper focuses on a different question: given a cache of past interactions, when should the system return cached responses while enforcing a user-defined error rate bound and being efficient. We do not make claims about privacy guarantees, and vCache can potentially be combined with standard data-governance mechanisms (per-tenant caches, access control, retention policies, PII filtering) that determine the actual leakage risk in practice. A systematic study of how semantic caching interacts with privacy (e.g., how cache partitioning, retention, and error-rate control jointly affect leakage) is an interesting direction for future work.
>
> ## SemCacheLMArena
> The SemCacheLMArena’s construction is described in detail in Appendix F.2: we first sample seed prompts from LMArena and then generate multiple paraphrases per seed, so high cache hit rates reflect semantic redundancy (paraphrase clusters), not “testing cases” leaking into evaluation. For SemCacheClassification and SemCacheSearchQueries, prompts are sampled independently from their underlying datasets (without paraphrase expansion), so their cache hit rates are not driven by LMArena-style prompts.
>
> Although we conducted a thorough and thoughtful review, please let us know if we missed anything relevant. We encourage the reviewer to update their score if they feel their concerns have been fully resolved.

---

### Official Review · Reviewer_e1XS · 2025-10-31

**Soundness:** 3
**Presentation:** 3
**Contribution:** 3
**Rating:** 6
**Confidence:** 2

**Summary:**

This paper proposes vCache, a caching system that optimizes GPT-Cache and dynamically returned the cached responses subjected to the pre-defined error rate. Compared to the previous works that relies on a static threshold for all incoming queries and all cached responses, this paper adopts online learning to estimate a threshold for each cached response. To help evaluate the performance of GPT-cache style caching systems, the author generates three benchmarks to help evaluate the performance. Abundant empirical experiments on different models show the effectiveness of this method.

**Strengths:**

- This paper tackles on an important topic of GPT-cache and provides unique perspectiives on how to optimize this problem. Static threshold is a fundmental problem that prevents GPT-cache for better selecting cached responses.
- The evaluation is holistic and abundant. The author also presents large-scale benchmarks for helping research in this field.
- Author also presents good theoretical guarantees for showing the effectiveness of this method,

**Weaknesses:**

- Dataset Generation neglects no match scenario. I checked the appendix about how the data is generated. For instance, for the SemCacheLMArena, 1 to 23 similar prompts will be generated. In reality, there could be many prompts where no similar answer in the cache can be fetched directly. I would suggest adding many prompts where no similar one variants are included should be helpful.
- Though the experiments regarding error rate is abundant, more analysis of latencies and throughputs should be added.

**Questions:**

- For creating the dataset SemCacheLMArena, how to ensure the sampled prompts are not really distinct?
- Where do you define? I guess it's time per response?

---

> ### Author Response · Authors · 2025-11-21
> **Response to yfEW**
>
> We thank the reviewers for their careful reading and constructive feedback, and we address their main concerns point by point below.
>
> ## A new dataset with no match cases
> Thank you for highlighting the importance of no-match scenarios. Motivated by this, we introduce the SemCacheCombo benchmark, which combines prompts with potential cache hits and prompts that have no semantically similar response in the dataset. We describe the construction in Appendix F.4, release the dataset on HuggingFace, and report results in Appendix D.2. On this benchmark, vCache consistently outperforms the state-of-the-art GPTCache baseline, achieving up to 12.5× higher cache hit rates while satisfying the user-defined error bound across all settings.
>
> ## Experiments on latency and throughput
> We agree that latency and throughput are important metrics, and we already evaluate them at both the system and component levels. At the end-to-end level, Figures 5, 8, 10, and 12 report average latency jointly with error and cache hit rate across configurations, and Table 2 measures the full vLLM+vCache system on 45k SemCacheClassification queries, showing a reduction in total runtime from 240 to 160 minutes on the same hardware and workload. At the component level, Figure 13 isolates the overhead of computing the exploration probability $\tau$ (effectively constant and below 1.5, ms per update), and Figure 14 compares embedding-generation latency to LLM inference. For a fixed hardware configuration and fixed number of requests, throughput is the inverse of average per-request latency.
>
> ## SemCacheLMArena semantics
> For SemCacheLMArena, we do not cluster unrelated prompts post-hoc; instead, each equivalence class is explicitly constructed around a single human-written LM-Arena query. Concretely, we first sample 3,500 distinct prompts from the LM-Arena human preference dataset and treat each as a class anchor; for every anchor, we then ask GPT-4.1-nano to generate multiple paraphrases under an explicit “rewrite this query without changing its meaning or desired answer” instruction, yielding 1–23 variants per class and 60,000 prompts in total. Because all variants are generated conditionally on the same seed prompt and constrained to preserve intent, they differ primarily in surface form (wording, order, and local detail) while still admitting the same valid responses, which is exactly the setting semantic caches target. We assign a class ID to each prompt and define a cache hit as correct if the retrieved response comes from the same class, so any occasional drift where GPT-4.1-nano produces a genuinely different question is treated as label noise that makes the benchmark slightly harder.
>
> ## “Where do you define? I guess it's time per response?”
> Thank you for the comment. Could you please clarify which quantity you are referring to?
>
> Although we conducted a thorough and thoughtful review, please let us know if we missed anything relevant. We encourage the reviewer to update their score if they feel their concerns have been fully resolved.

---

### Official Review · Reviewer_yfEW · 2025-11-01

**Soundness:** 3
**Presentation:** 3
**Contribution:** 2
**Rating:** 6
**Confidence:** 4

**Summary:**

The paper proposes vCache, a novel semantic caching system for LLMs that provides user-defined error rate guarantees while learning embedding-specific thresholds online. The key innovation is replacing static global thresholds with dynamic, per-embedding thresholds estimated through an online learning algorithm that models the probability of correctness using sigmoid functions. The method is evaluated on three benchmarks across different embedding models and LLMs, demonstrating superior performance compared to static threshold baselines.

**Strengths:**

- The paper addresses a practical challenge of the semantic prompt caching
- The proposed method is well-motivated

**Weaknesses:**

- The writing is a bit repetitive and could be streamlined

**Questions:**

Thank you for your submission. I appreciate the motivation behind providing formal correctness guarantees for semantic caching systems, which is indeed a significant limitation of existing approaches. The experimental validation across multiple benchmarks and the introduction of new evaluation datasets are valuable contributions. However, several aspects of the paper require clarification and the technical approach raises some concerns:

- The sigmoid modeling assumption (Equation 9) is quite strong but not well justified. Why should the relationship between similarity and correctness follow a sigmoid specifically? Have you experimented with other parametric families or non-parametric approaches?
- The confidence band computation for parameters t and γ (mentioned in Section 4.2 and relegated to Appendix C) is crucial for the guarantees but insufficiently explained in the main text. How sensitive are the guarantees to the choice of confidence level ε?
- Algorithm 2 shows that τ is computed by minimizing over ε ∈ [0,1], but this seems computationally expensive for online inference. What is the actual computational overhead of this optimization step?
- The paper claims vCache "consistently meets the specified error bounds" but Figure 4 shows the actual error rate is noticeably below the specified δ. This suggests the method might be overly conservative, potentially sacrificing cache hit rate for unnecessary safety margins. Can you quantify this conservatism?
- The comparison with GPTCache is somewhat unfair since GPTCache doesn't attempt to provide error guarantees. A more relevant baseline would be other adaptive thresholding methods from the retrieval literature adapted to this setting.
- The evaluation focuses on relatively simple benchmarks (classification, search queries). How does vCache perform on more complex scenarios like multi-turn conversations or reasoning tasks where semantic similarity becomes more nuanced?
- The i.i.d. assumption for incoming prompts is quite restrictive in practice. Real-world query distributions often exhibit temporal correlations, user-specific patterns, and concept drift. How robust is vCache to violations of this assumption?
- Figure 3's motivation is compelling, but the connection to the proposed solution could be clearer. It shows the problem but doesn't intuitively explain why sigmoid modeling would solve it.

---

> ### Author Response · Authors · 2025-11-21
> **Response to yfEW (Part I)**
>
> We thank the reviewers for their careful reading and constructive feedback, and we address their main concerns point by point below.
>
> ## Justification for Sigmoid function and Figure 3
> We thank the reviewer for this comment and have added Appendix H to clarify and empirically justify the modeling choice in Eq. (9). There, we analyze SemBenchmarkLmArena with text-embedding-3-small and GPT-4.1-nano, and show that both per-embedding and aggregate estimates of $Pr⁡(c(x) = 1 | s(x))$ are monotone and S-shaped in the similarity s(x), which matches the behavior captured by the sigmoid family used for $\mathcal{L}(s(x), t, \gamma)$.
>
> To clarify the role of Figure 3: it illustrates that a single global similarity threshold cannot capture the embedding-specific relationship between similarity and correctness, while Eq. (9) (together with Algorithm 2) replaces this with an embedding-specific, smooth curve $\mathcal{L}(s(x), t, \gamma)$ that is consistent with the empirical shapes shown in Appendix H.
>
> ## Confidence band explanation and sensitivity analysis
> Our use of $\varepsilon$ is analogous to standard concentration bounds (e.g., Chernoff bounds), which provide a whole family of high-probability guarantees indexed by a deviation parameter. For each $\varepsilon$, Lemma C.2 yields a valid finite-sample guarantee on $Pr⁡(c(x) = 1 |  \mathcal{D}, x)$ just like Chernoff yields a valid deviation inequality for each $\varepsilon$. Algorithm 2 then selects the tightest resulting bound $\tau'(\varepsilon)$ by minimizing over $\varepsilon$; taking the minimum over individually valid upper bounds preserves correctness, exactly as in classical concentration-inequality practice. Therefore, our $\delta$-guarantee is not tied to a specific ε and holds uniformly over all $\varepsilon$ considered.
>
> ## Computational overhead analysis for $\tau$ in Algorithm 2
> We conduct a latency overhead analysis for the $\tau$ computation and add the results to Appendix D.4. In short, the latency remains constant and below 1.5ms.
>
> ## Quantify gap between error bound and actual performance
> We agree that the gap between the specified error rate bound $\delta$ and the realized error rate in Figure 4 is important to understand. vCache is designed as a one-sided, high-probability guarantee: we control the upper tail of the error curve by using an upper confidence band when choosing thresholds. As long as there is residual statistical uncertainty, this construction is intentionally conservative and will tend to produce realized error rates below $\delta$; this is expected behaviour for a method that aims to guarantee error ≤ $\delta$ uniformly over time and across many embeddings, rather than matching $\delta$ exactly on average.
>
> ## Other baselines with guarantees from IR
> We agree that GPTCache does not aim to provide error guarantees, underlining its limitation. We use it as the most widely adopted semantic caching system for LLMs and compare on common outcome metrics (cache hit rate, error rate, and roc), while vCache additionally provides user-defined error bounds. To address the reviewer’s point about adaptive thresholding, we already include several dynamic-threshold baselines beyond GPTCache: GD (global dynamic threshold), LS (per-embedding local static threshold), and LD1/LD2 (per-embedding local dynamic thresholds without guarantees), as summarized in Table 3. These baselines explicitly probe the design space of global vs.\ per-embedding thresholds and static vs.\ dynamic adaptation, and vCache (LD3) outperforms them while also enforcing the user-specified error rate.
> We are not aware of existing methods in the information retrieval literature that both (i) operate in the semantic caching setting and (ii) provide user-definable error-rate guarantees for reusing generated LLM responses. Adapting retrieval-oriented thresholding schemes to this setting (e.g., defining labels, calibration targets, and online update rules) would itself constitute substantial new work beyond the scope of this paper. That said, if the reviewer has a specific adaptive-thresholding method in mind that can be instantiated in our semantic caching setup, we would be happy to include it as an additional baseline in a revised version.

---

> ### Author Response · Authors · 2025-11-21
> **Response to yfEW (Part II)**
>
> ## Performance on more complex benchmark
> Our goal in this work is not to design a semantic caching system for long-context, multi-turn dialogue, but to improve existing semantic caches in the most common setting: single-turn interactions with short to medium context (e.g., search queries, classification workloads, and one-turn chatbot prompts). In this regime, vCache contributes (i) user-defined error-rate bounds and (ii) embedding-specific thresholds learned online, yielding a better accuracy-latency trade-off than the state-of-the-art. Our four benchmarks are chosen to reflect this scope. Within this setting, vCache consistently respects the user-defined error bound while increasing cache hit effectiveness. Extending semantic caching to full multi-turn conversations and complex reasoning introduces additional challenges (e.g., evolving dialogue state) that are orthogonal to our focus on threshold learning and guarantees. We view the extension to full multi-turn conversations as important future work.
>
> ## IID
> While it is true that the vCache algorithm depends on the IID assumption, as is true for many statistical recipes, we expect the behavior of vCache to gradually deteriorate when introduced with drift, being generally robust to small drifts and breaking with extreme drifts. We are not aware of any prompt benchmarks that are time-stamped and contain drifts. It would be of immense help if the reviewer could point to any such benchmark/data that can be used to create such a benchmark. We can then test the robustness of vCache.
>
> Although we conducted a thorough and thoughtful review, please let us know if we missed anything relevant. We encourage the reviewer to update their score if they feel their concerns have been fully resolved.

---

### Official Review · Reviewer_5kcm · 2025-11-01

**Soundness:** 3
**Presentation:** 2
**Contribution:** 3
**Rating:** 6
**Confidence:** 3

**Summary:**

This paper proposes vCache, a semantic LLM caching method that guarantees a user-defined error rate, based on the previous work of GPTCache, a semantic LLM caching method to optimize LLM cost and latency. This work contributes to the domain of LLM caching by introducing per-embedding dynamic threshold, which was improved from the static threshold of GPTCache.

**Strengths:**

1.	This study algorithmically improves from the existing threshold-based retrieval methods such as GPTCache using a novel approach. It theoretically guarantees a desired error rate and demonstrates improved performance across several metrics by introducing a verified semantic cache.
2.	As an online learning algorithm, the proposed method achieves strong results without fine-tuning the embedding model, thus requiring no additional training.
3.	To validate the effectiveness of the proposed methodology and foster future research, the authors have constructed and publicly released three new benchmark datasets (SemCacheClassification, SemCacheLMArena, and SemCacheSearchQueries) that reflect real-world caching scenarios.

**Weaknesses:**

1.	Since vCache is still a semantic caching technique, there is an insufficient evaluation of its effectiveness concerning the capability and performance across the underlying embedding model (e.g., BERT). A comparative analysis using various embedding models is needed to substantiate the "Model-Agnostic" claim, which currently lacks sufficient analysis.
2.	While the paper discusses the trade-off between accuracy and cost, the benchmark results show a trade-off between Cache Hit Rate and Error Rate when compared to GPTCache. For instance, in Appendix D.5, the best-case for GPTCache (GS) shows a 5.2% error rate with a 67% hit rate, whereas vCache (LD3) achieves a 2.0% error rate with a 54% hit rate. Although vCache has the distinct advantage of reliably guaranteeing a user-defined error rate (e.g., 2.0%), the lack of in-depth analysis on this trade-off makes it difficult to conclude its superiority across all scenarios. Combined with Weakness 1, this property raises questions about whether the improvement of vCache against GPTCache is marginal or not.
3.	As mentioned in the limitations section, the evaluation relies on an LLM-as-a-judge for benchmarks except SemCacheClassification, which has a clearly defined correctness criterion.

**Questions:**

•	Were comparative experiments conducted with different embedding models? An analysis of the relationship between the performance of the embedding model and the performance gains from vCache would better substantiate the "Model-Agnostic" claim. Beyond the GteLarge and E5-large models presented, were experiments with other BERT models—such as multilingual variants or those employing improved techniques—considered?
•	When a new embedding is added to the cache, how many observations are required for vCache to learn a stable threshold? I am curious about the analysis of performance degradation during the initial learning phase (the cold-start problem).

---

> ### Author Response · Authors · 2025-11-21
> **Response to 5kcm (Part I)**
>
> We thank the reviewers for their careful reading and constructive feedback, and we address their main concerns point-by-point below.
>
> ## Evaluating a third embedding model
> To further support our claim that vCache is embedding-model agnostic, we added experiments using OpenAI’s text-embedding-3-small model, in addition to GTE-large-en-v1 and E5-large-v2. Across all three embeddings, vCache outperforms GPTCache while satisfying the user-defined error bounds; with OpenAI’s text-embedding-3-small in particular, vCache achieves up to 26x lower error rates than GPTCache. The detailed results are reported in Appendix D.3.
>
> ## Trade-off between accuracy and cost
> We agree that the accuracy–cost trade-off is important. In our setting, with a fixed LLM and stable token lengths, the monetary cost is linear in the number of LLM calls, so the cache hit rate is proportional to cost savings. This is why the main results are presented as cache hit rate vs. error rate.
>
> To make this connection explicit, we additionally ran a cost analysis using current OpenAI pricing. On our benchmark without caching, a full run costs $11.45 with GPT-4.1-nano and 8.13 with GPT-4o-mini. At a fixed 2% error rate, the new experiments show that vCache achieves a 55% cache hit rate, compared to 20% for GPTCache.
>
> This translates into the following reduction in LLM-generation cost:
>
> | Model | No Cache | GPTCache | vCache | VCache vs. GPTCache |
> |-------|----------|----------|--------|---------------------|
> | GPT-4.1 nano | $11.45 | $9.16 | $5.15 | 1.8x |
> | GPT-4o mini | $18.13 | $14.50 | $8.16 | 1.8x |
>
> The additional embedding cost for text-embedding-3-small/large is $0.06 – 0.37 per run. Overall, at the same 2% error target, vCache delivers 1.8x higher cost savings than GPTCache.
>
> ## vCache only marginally better than GPTCache?
> We agree that understanding the full trade-off between cache hit rate and error rate is crucial. The example in Appendix D.5 (GPTCache: 5.2% error, 67% hit; vCache: 2.0% error, 54% hit) reflects only a single operating point for each method.
>
> To analyze this trade-off more systematically, we explicitly evaluate the Pareto frontier over all feasible thresholds and error bounds (Figures 5 and 8, and now also Figures 10 and 12). On SemCacheLMArena and SemCacheSearchQueries (Figures 5 and 8), vCache matches or exceeds GPTCache across the frontier: for any target error rate, vCache achieves comparable or higher hit rates while respecting the user-defined error bound. On SemCacheCombo and with OpenAI’s ext-embedding-3-small embeddings (Figures 10 and 12), vCache substantially exceeds GPTCache, achieving up to **12.5× higher cache hit rates** at the same error level and up to **26× lower error at comparable hit rates**.
>
> These results indicate that the improvement is not marginal when viewed over the entire trade-off surface. Moreover, vCache is the only method that allows users to choose an error bound, whereas GPTCache’s error rate is uncontrolled and may increase over time (Figures 4, 5, 7, 9).

---

> > ### Author Response · Authors · 2025-11-21
> > **Response to 5kcm (Part II)**
> >
> > ## LLM-as-a-judge
> > We agree that the choice of the evaluation oracle is important. In our setting, the judge is not part of the vCache algorithm; it is only used to obtain binary correctness labels $c(x) \in {0,1}$ for long-form responses. Our theoretical guarantee and decision rule only assume access to such labels and are agnostic to whether they come from humans, heuristics, or an LLM.
> >
> > To mitigate dependence on a specific judge, we evaluate vCache under three regimes:
> >
> > 1. SemCacheClassification benchmark, where correctness is determined by exact string matching over a fixed label set;
> > 2. SemCacheLMArena and SemCacheSearchQueries benchmarks, where equivalence of mid-form responses is assessed with a standard LLM-as-a-judge protocol; and
> > 3. SemCacheCombo, which mixes both types of workloads (newly added).
> >
> > Across all regimes, vCache satisfies the user-defined error bound and outperforms state-of-the-art baselines.
> >
> > For open-ended prompts, exact matching is infeasible, and LLM-as-a-judge has become standard practice, with prior work (see “Judging LLM-as-a-Judge with MT-Bench and Chatbot Arena”) showing high agreement with human annotations.
> >
> > ## vCache algorithm convergence speed
> > It is important to emphasize that vCache does not rely on point estimates. Instead, it takes a Bayesian approach, maintaining a distribution over the correctness function rather than committing to a single value. This has a crucial effect: when the number of observations is small, the posterior distribution remains diffuse, reflecting high uncertainty. As a result, vCache naturally behaves more conservatively in low-data regimes and avoids overly confident, error-prone decisions.
> > At the same time, the posterior distribution concentrates at standard statistical rates. In particular, vCache fits a local logistic model whose parameters ($(\hat{t}_y, \hat{\gamma}_y)$, and thus any embedding-specific threshold and exploration probability, converge at the standard parametric rate $\mathcal{O}(1/\sqrt{n_y})$ in the number of explored queries $n_y$ with nn(x) = y (see Appendix G).
> >
> > vCache’s confidence-band construction automatically widens for small $n_y$​ , inducing higher exploration probabilities and thereby preserving the global $\delta$-accuracy guarantee even in the cold-start regime.

---

### Author Response · Authors · 2025-11-21
**Response to all reviewers**

We thank all reviewers for their time, constructive comments, and positive evaluations. Below we summarize the main clarifications, new experiments, and additional evidence added in the rebuttal and revised manuscript.

- **New benchmark SemCacheCombo** [Appendix D.2]. We add SemCacheCombo, which combines SemCacheClassification and SemCacheLMArena to explicitly cover workloads with and without cache hits. On this benchmark, vCache consistently outperforms state-of-the-art baselines, achieving up to 12.5× higher cache hit rates while satisfying the user-defined error-rate bound.
- **Additional embedding model** [Appendix D3]. We evaluate a third embedding model, text-embedding-3-small, alongside GPT-4.1-nano, complementing the two embedding models already studied. vCache outperforms the GPTCache baselines with up to 26× lower error rates while respecting the user-specified error bound.
- **Overhead of $\tau$ computation** [Appendix D.4]. We measure the latency of computing 𝜏 and show that, across all sample sizes, the per-query overhead remains below 1.5 ms, indicating that the computation of $\tau$ is negligible compared to LLM latency.
- **Justification of the sigmoid model** [Appendix H.]. We add experiments analyzing $Pr ⁡ (c(x) = 1 | s(x))$ as a function of similarity, both per-embedding and aggregated across embeddings, and show that the empirical curves are monotone and S-shaped, supporting our sigmoid-based model in Eq. (9).

---

### Meta-Review · Area_Chair_eCB8 · 2026-01-06

**Summary:**

The paper presents a semantic caching approach that estimates per-prompt thresholds, thus avoiding the problem of highly overlapping similarity distributions for correct and incorrect responses. This is achieved by modelling the probability of cache correctness as a function of embedding similarity.

Reviewers were generally positive about the work, but raised a few concerns:
- **Empirical versus theoretical error bound**. Multiple reviewers expressed confusion that the results in Figure 4 are significantly more conservative than the bound predicted by the theory.
- **Reliance on LLM-as-a-judge**. Multiple reviewers noted that the method relies on LLM-as-a-judge labels for many settings, which requires a separate LLM inference call. This raised the concern of error propagation, but also increased inference cost.
- **Sigmoid modelling assumption**. Multiple reviewers noted the assumption of a sigmoid form for the cache correctness probability $\Pr(c(x) = 1 \mid x, \mathcal{D})$ appears to be strong, with unclear justification.
- **Overhead of $\tau$ computation**. One reviewer that the algorithm requires a search over $\tau$ values, which could add inference cost.
- **Fairness of GPTCache comparison**. Multiple reviewers noted that the GPTCache comparison may not be fully fair, as it appears to operate with different hit rate versus quality tradeoffs, and GPTCache was not designed with error guarantees.
- **Role of embedding model**. One reviewer noted that the choice of embedding model should be ablated, as this critically determines the notion of semantic similarity amongst prompts.
- **Restriction to simple tasks**. Multiple reviewers noted that the results are limited to relatively simple tasks, and do not consider, e.g., multi-turn or long-context settings.
- **Privacy concerns**. One reviewer noted that caching strategies introduce issues with data leakage, which could hinder wider adoption.
- **Writing quality**. One reviewer noted that there is scope for streamlining of the writing.

**Reviewer Concerns:**

- **Empirical versus theoretical error bound**. The authors clarified that the bound in Equation 12 is a one-sided, high-probability guarantee, which holds over the prompts drawn at test time.
  - *Mostly addressed*. The authors' response helps explain the trend in Figure 4. Such an explanation would be worth adding to the paper.
- **Reliance on LLM-as-a-judge**. The authors acknowledged that some feedback mechanism is needed, but noted that this could be in the form of application logic or offline labels. They also noted that these labels were not required for every request, thus limiting the computational burden. They also reported results on SemCacheCombo, which uses a combination of exact match and LLM-as-a-judge labels, which shows favorable results.
  - *Mostly addressed*. The authors' response is reasonable, and describes an shared caveat for any semantic caching method. The new SemCacheCombo results help strengthen the main claims.
- **Sigmoid modelling assumption**. The authors added in Appendix H an empirical analysis of the correctness probability as the similarity is varied, finding it to vary monotonically with a shape argued to be sigmoidal.
  - *Mostly addressed*. The new analysis is welcome. It would be clearer if Figure 18 could overlay a best-fit sigmoid curve (with suitable temperature and offset parameter). More generally, it seems that the key claims are that (a) the correctness is a monotone function of the similarity, (b) the sigmoid is a reasonable choice of monotone function. Assuming (a), (b) seems fairly reasonable a-priori, and is further evidenced by Figure 18. For (a), some more intuitive discussion would be advisable. The authors may also see about converting the density plots of Figure 3 into a plot of the density ratio $p( s \mid c(x) = 1 ) / p( s \mid c(x) = 0 )$.
- **Overhead of $\tau$ computation**. The authors presented a calculation of the cost of computing $\tau$, finding it to be negligible (1.5 ms)
  - *Mostly addressed*.
- **Fairness of GPTCache comparison**. The authors presented new results comparing vCache to GPTCache with a fixed error rate of 2%, and converting the hit rate into dollar amounts based on the OpenAI API pricing. These show ~80% reduction in dollary cost for a fixed error rate. The authors also argued that the lack of error guarantees underscore the limits of GPTCache, while noting that repurposing existing methods from the retrieval literature would be a significant contribution by itself.
  - *Mostly addressed*. The new results support the main claim well. It is natural to wonder how the results are for a _range_ of different error rates (i.e., can you trace out a more complete error rate versus hit rate / dollar curve?).
- **Role of embedding model**. The authors provided additional results with the text-embedding-3-small embedding model.
  - *Mostly addressed*. The additional results lend credence to the claim that the vCache method offers gains not tied to one particular notion of semantic similarity.
- **Restriction to simple tasks**. The authors acknowledged that the scope of tasks considered is not exhaustive, but argued that their setup represents the common case for semantic caching. They also argued that settings such as non-IID prompts would be of interest, but lack any standard benchmark.
  - *Mostly addressed*. We agree with the authors that the tasks considered are reasonable for the assessment of advances in semantic caching. Settings such as multi-turn dialogue would require interesting but orthogonal modelling details.
- **Privacy concerns**. The authors noted that any privacy preserving mechanism can be applied on top of the approach (or indeed, any semantic caching approach).
  - *Mostly addressed*. We agree with the authors that while the issue is important, it is orthogonal to the focus of the paper.
- **Writing quality**. The authors did not explicitly discuss this point.
  - *Not addressed*. We would encourage the authors to consider edits to streamline the presentation. From our reading, we make a few notes.
    - Section 3 is slightly jarring in that it begins purporting to present a general overview of semantic caching, but then interjects with mention of vCache (which has not been formally introduced at this point).
    - Consider using $\mathrm{sim}$ rather than $sim$, $\mathrm{nn}$ rather than $nn$.
    - Equation 5, the left equation should use $(s(x), c(x))$, not $s(x), c(x)$.
    - Following Equation 6, both $\Pr_{\rm explore}( x \mid \mathcal{D} )$ and $\Pr( {\rm explore} \mid x, \mathcal{D} )$ are used; the latter seems to be the intended notation.
    - Equation 8, not initially clear whether $\tau$ is intended to be defined to equal the RHS.

**Reviewer Scores:**

- **5kcm**: as their comments were mostly addressed, we think it likely the score would remain at 6, with some chance of increasing to 8.
- **e1XS**: as their comments were mostly addressed, we think it likely the score would be a 6, with some chance of increasing to 8.
- **sFQ7**: as some of the reviewers' critiques were addressed, we think it possible the score would increase to 6. However, as some points may not have been addressed to full satisfaction, we think it also possible that the score would remain at 4.
- **yfEW**: as the reviewer raised a number of detailed comments, a few of which may not have been addressed to full satisfaction, we think it likely that the score would remain at 6.

---

### Decision · Program_Chairs · 2026-01-26

Accept (Poster)